# Extracellular vesicles from neurons promote neural induction of stem cells through cyclin D1

Lu Song⊙, Xinran Tian, and Randy Schekman⊙

**Extracellular vesicles (EVs) are thought to mediate the transport of proteins and RNAs involved in intercellular communication. Here, we show dynamic changes in the buoyant density and abundance of EVs that are secreted by PC12 cells stimulated with nerve growth factor (NGF), N2A cells treated with retinoic acid to induce neural differentiation, and mouse embryonic stem cells (mESCs) differentiated into neuronal cells. EVs secreted from in vitro differentiated cells promote neural induction of mESCs. Cyclin D1 enriched within the EVs derived from differentiated neuronal cells contributes to this induction. EVs purified from cells overexpressing cyclin D1 are more potent in neural induction of mESC cells. Depletion of cyclin D1 from the EVs reduced the neural induction effect. Our results suggest that EVs regulate neural development through sorting of cyclin D1.**

## Introduction

Intercellular communication involves either direct contact between neighboring cells or indirect interaction via secreted factors including extracellular vesicles (EVs; Hessvik and Llorente, 2018). Cells release two subtypes of EVs according to their cellular membrane origin: plasma membrane–derived EVs and endosome-derived exosomes (French et al., 2017). Plasma membrane–budded vesicles range from 30 to 1,000 nm in diameter, whereas exosomes range from 30 to 150 nm in diameter. Intralumenal vesicles that accumulate within multivesicular bodies (MVBs) are secreted as exosomal EVs to the extracellular environment upon fusion of MVBs with the plasma membrane. EVs are enclosed by a lipid bilayer containing transmembrane proteins, luminal cytosolic proteins, and nucleic acids (Raposo and Stoorvogel, 2013). Much interest has focused on the wide distribution of EVs in all biological fluids and their potential to trigger intercellular exchange of effector molecules, which may allow secretion of cells to modulate gene expression in target cells and tissues (Mulcahy et al., 2014).

The lipid membrane of EVs ensures the stability of lumenal cargo as vesicles circulate in the extracellular space, potentially over long distances. The small size of EVs helps overcome various biological barriers, including the blood–brain barrier (van Niel et al., 2018). In the brain, EVs represent an ideal vehicle for intercellular transfer of information from neurons and glia to both neighboring and distal cells (Budnik et al., 2016). Multiple cell types in the mature central nervous system release EVs, including neurons, astrocytes, and oligodendrocytes (Pascual et al., 2020). Neuronally secreted EVs could control synaptic

plasticity and enhance the removal of degenerative neurites after internalization by microglial cells (Bahrini et al., 2015). EVs secreted by astrocytes display neuroprotective activity that is critical for neuronal cell survival (Verkhratsky et al., 2016). Oligodendrocytes secrete EVs that are endocytosed by neurons and increase their viability (Krämer-Albers et al., 2007). In addition to studies on EVs in the mature central nervous system, recent studies provide evidence that EVs from newly differentiated neural cells promote neurogenesis (Sharma et al., 2019).

In neural development, cell fate determination is tightly controlled by stepwise commitment, including neural induction and neurogenesis (Grow, 2018). Pluripotent stem cells convert into neural ectoderm progenitors, after which neural precursors further differentiate into nerve cells of defined function (Muñoz-Sanjuán and Brivanlou, 2002). Although EVs have been suggested to facilitate the later neurogenesis events, little is known about the role of EVs during the early stage of neural fate conversion. Furthermore, clear and direct evidence of such roles is lacking, as the studies thus far have relied on highly impure, crude preparations of sedimented particles (Sharma et al., 2019).

Much of the literature on the proposed function of EVs has relied on differential sedimentation of slowly sedimenting or crudely precipitated particles obtained from culture medium or other fluids. This method does not separate large shedding microvesicles from small exosomal-like EVs, nor does it remove protein aggregates or ribonucleoprotein particles (Konoshenko et al., 2018). Further resolution can be achieved by a series of ultracentrifugation and density gradient centrifugation steps

..............................................................................................................................................................................
Department of Molecular and Cell Biology, Howard Hughes Medical Institute, University of California, Berkeley, Berkeley, CA.

Correspondence to Randy Schekman: schekman@berkeley.edu.

that remove cells, cell debris, microvesicles, and protein aggregates (Shurtleff et al., 2016, 2018; Temoche-Diaz et al., 2019).

To investigate the role of purified EVs during neural development, we used buoyant density flotation to isolate EVs from nerve growth factor (NGF)–induced PC12 cells and retinoic acid (RA)–induced neuro 2A (N2A) cells. We then examined the effect of purified vesicles on mouse embryonic stem cells (mESCs) and found that the neuronal EVs accelerate aspects of mESC neural induction. We further demonstrated that a cell cycle–related factor, cyclin D1, was enriched within EVs derived from differentiated cells. Compared with the EVs from untreated cells, those purified from cells overexpressing cyclin D1 enhanced neural lineage gene expression in mESCs. Conversely, EVs from cyclin D1 knockout cells did not stimulate neural induction of mESCs. The chaperone protein Hsc70 facilitated packaging of cyclin D1 into EVs. Our results suggest that EVs contribute to neural fate determination through sorting of cyclin D1.

## Results

### Dynamic changes of EVs secreted by neuronal differentiated PC12 cells

We compared EVs secreted from undifferentiated neuronal progenitor-like cells with those secreted by differentiated neuronal cells. In an initial approach, we used PC12 cells, which differentiate after treatment with NGF. After 9 d of NGF treatment, long neurite extensions consistent with a neuronal fate were observed (Fig. 1 A). Consistently, quantitative PCR (qPCR) results showed that the neuronal marker genes, *Tuj1* and *Tau*, were up-regulated concomitant with neurite extension. We tested two NGF concentrations and found that both elicited similar levels of expression of neuronal marker genes (Fig. 1 B). As a result, the lower dosage of NGF, 50 ng/ml, was used in all subsequent experiments.

Next, to investigate if the protein composition of EVs changes during PC12 cell neuronal differentiation, we collected samples at different time points after adding NGF. EVs were obtained from the culture medium by differential centrifugation to concentrate small particles, followed by equilibrium sedimentation in an iodixanol linear density gradient. Each fraction was collected from top to bottom for further analysis (Fig. 1 C). Flot2, a lipid scaffolding protein, showed no dramatic change in the expression of EVs (in buoyant densities ranging from 1.06 to 1.23 g/ml) from cells treated with NGF for 0, 3, 6, and 9 d. Thus, vesicles marked by this reporter protein appeared not to vary during the differentiation process (Fig. 1, D and E). However, the EV marker CD9, a tetraspanin enriched in MVBs, showed an obvious shift in the buoyant density peak in EVs from cells treated for 3 d. The prominent peaks changed from fractions corresponding to densities of ~1.08 g/ml to ~1.13 g/ml at the onset of differentiation. During the following 6 d, the CD9-containing EVs showed a broader density distribution (Fig. 1, D and E). Consistently, heat shock chaperone Hsc70, which is also a well-known EV marker, displayed a similar expression pattern during differentiation (Fig. 1, D and E). Furthermore, the size of vesicles (in even-numbered fractions) analyzed with a

NanoSight particle tracking device showed no dramatic difference between subfractions. In contrast, the protein concentration appeared to increase gradually in fractions containing vesicles from low to high buoyant density (Fig. S1, A and B). These results indicated that the physical properties of CD9-containing EVs changed over the course of differentiation.

### EV production increased during neuronal differentiation

EVs secreted from NGF-induced PC12 cells were further characterized by independent isolation using differential centrifugation and sucrose gradient buoyant density flotation to obtain membranes at the 20/40% interface, with a density corresponding to ~1.12 g/ml (Fig. 2 A). Membranes were sedimented, washed, and resuspended to assess vesicle morphology. Negative-stain EM images displayed a characteristic cup-shaped, collapsed appearance in all four groups of EV preparations (Fig. 2 B).

EV particle size and number were analyzed by NanoSight. The average vesicle diameter varied slightly from 126 ± 2 to 134 ± 2 nm after neuronal differentiation (Fig. 2 C). Quantification documented an increase in EV production in PC12 cells during neuronal differentiation. EVs (~6 × 10^10; N6-EVs) were collected from 420 ml medium of NGF-induced 6-d cells, whereas ~2 × 10^10 EVs (PC12-EVs) were released into the same volume of medium from nondifferentiated PC12 cells (Fig. 2 C). After normalization to cell number, ~1,450 N6-EVs were secreted per differentiated PC12 cell, compared with ~500 per nondifferentiated PC12 cell (Fig. 2 D).

To determine if higher vesicle production was a common feature of differentiated neuronal cells, we employed another well-established neuronal differentiation system, the neuroblastoma N2A cell line. RA (10 μM) induced the differentiation of N2A ceclls as confirmed by cell morphology and expression of differentiation-specific marker, Neurogenin 2 (Ngn2; Fig. S2, A and B). Consistent with the observation of EVs from differentiated PC12 cells, after 6 d in RA-containing medium, ~7,000 EVs (RA6-EV) were produced/cell, whereas ~500 EVs/cell (N2A-EV) were secreted from nondifferentiated N2A cultures (Fig. 2 E).

To extend our observations to a more physiological source of neuronal cells, EVs were collected during mESC neural differentiation. Pluripotent ESCs at day 0 (ES D0) were cultured in suspension conditions for 8 d (ES D8) for conversion to embryonic bodies (EBs), which express neural progenitor markers. EBs were trypsinized in N2 medium for another 4 d (ES D12), during which time they differentiated into Tuj1+ neurons (Fig. S2, C and D). During 12 d of differentiation, ~400 EVs/cell (ES D12-EV) were released from neurons, whereas during 8 d of EB formation, ~200 EVs (ES D8-EV) were produced from each neural progenitor cell. As a control, we cultured mESCs in pluripotency-maintaining medium (N2B27 + 2i + leukemia inhibitory factor) for 2 d and found that ~80 EVs (ES D0-EV) were secreted/cell (Fig. 2 F).

We examined the expression of multiple EV marker proteins in sucrose gradient–purified vesicles released from 2 × 10^7 cells incubated in control and neuronal differentiation conditions. CD9, Hsc70, and Flot2, which were detected in the whole EV density profiles shown in Fig. 1, showed an approximately threefold increase in expression in the N6-EV compared with EVs from the untreated PC12 cells (PC12-EV). CD63, a tetraspanin

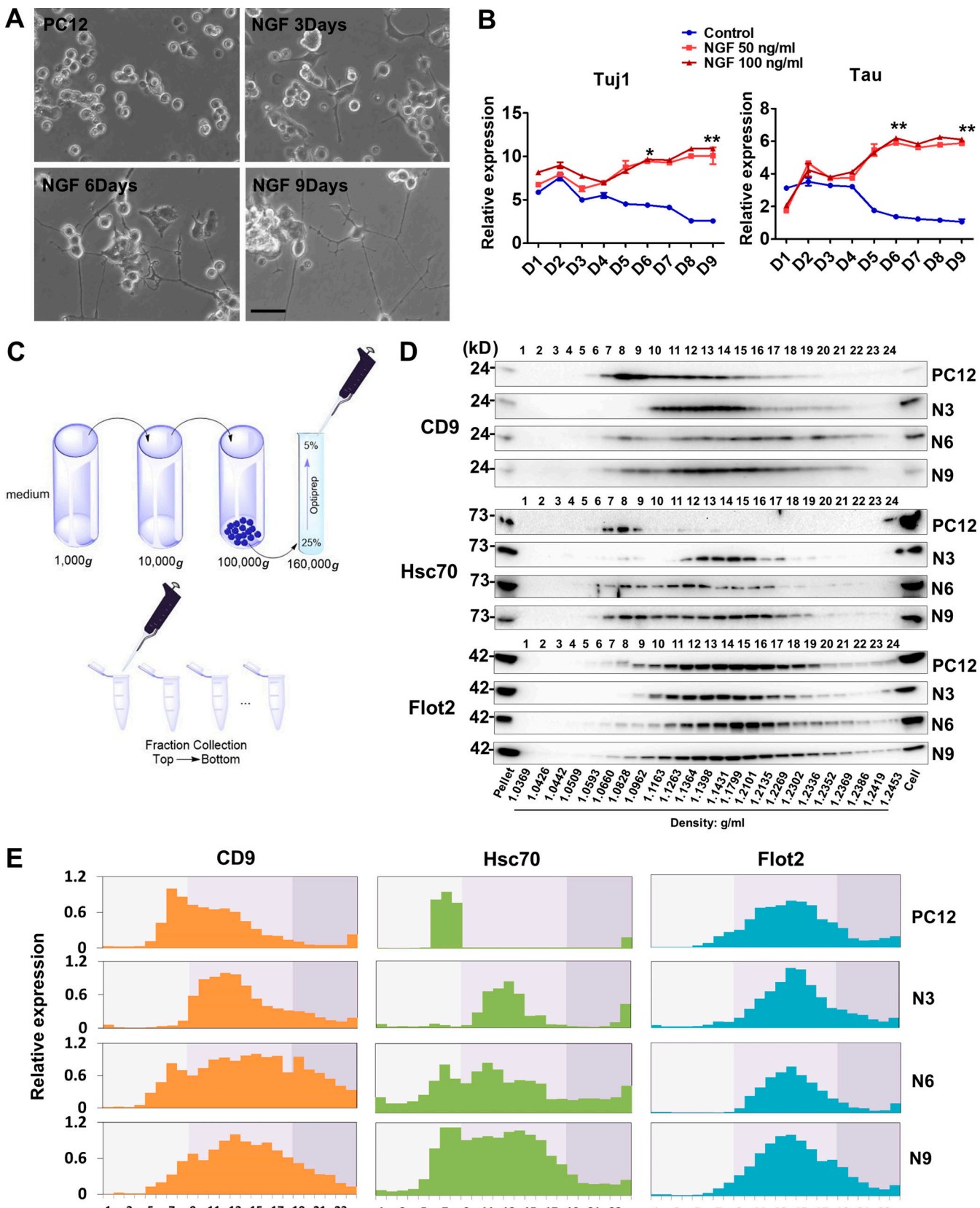

Figure 1. **EVs show different buoyant density distribution during PC12 neuronal differentiation. (A)** The cellular morphology of PC12 cells cultured in growth medium or low-serum medium with NGF (50 ng/ml) for 3, 6, and 9 d. Scale bars, 50 µm. **(B)** Expression profiling of *Tuj1* and *Tau* genes during neuronal differentiation of PC12 cells in low-serum medium without (Control) or with different doses of NGF (50 and 100 ng/ml). Expression was normalized to *Gapdh* in this and all others by qPCR analysis. Data plotted are from three independent experiments, each with triplicate qPCR reactions; error bars represent SD from

independent samples. The values represent the mean ± SD (*, P < 0.05; **, P < 0.01). **(C)** Schematic showing the fractionation of EVs. Differential ultra-centrifugation was followed by buoyant density flotation in a linear iodixanol gradient. **(D)** Immunoblot of EV markers of the iodixanol linear gradient fractions purified from PC12 cells untreated or treated with NGF for 3, 6, and 9 d (N3, N6, and N9). Pellet, 5% of 100,000 g vesicle pellet fraction was loaded in gel. Cell, whole-cell lysate (10 µg) was loaded in gel. The density of each fraction is indicated at the bottom. **(E)** Relative expression of CD9, Hsc70, and Flot2 from fraction 1 to fraction 24 shown in D. Data plotted represent the mean value from three independent experiments.

important for EV biogenesis, and Alix and Tsg101, essential components of the endosomal sorting complexes required for transport (aka ESCRT), were also up-regulated approximately threefold in the N6-EV versus PC12-EV (Fig. 2, G and H). EV production increased more dramatically in RA6-EV compared with undifferentiated N2A cells, with a ~14-fold increase in the same marker proteins (Fig. 2, I and J). These results suggested a possible role for the EVs produced during neuronal differentiation, at least as reflected in these cell lines.

**Purified EVs are taken up by mESCs**

Pluripotent mESCs may differentiate into various cell types, having the ability to commit to a specific lineage in response to external stimulation (Yu and Thomson, 2008). We used mESCs as the recipient cells to evaluate the influence of buoyant density gradient–purified EVs. mESCs were trypsinized to disperse single cells and cultured in serum-free N2B27 medium with EVs collected from differentiating PC12 cells. EVs were fluorescently tagged with the lipophilic membrane dye, PKH67, washed in PBS, and incubated for 24 h with mESCs. Labeled EVs from day 6 or 9 of NGF treatment appeared to be internalized in the recipient cells (Fig. 3, A and B).

We next tested if the uptake of a soluble, lumenal marker protein packaged into EVs could be internalized into recipient cells. For this purpose, we used lentivirus transfection to establish a GFP-overexpressing N2A cell line and then isolated EVs on a buoyant density gradient. The level of the EV marker CD9 and coincident GFP were detected by immunoblot in a linear range proportional to the number of EVs purified from RA-induced GFP-overexpressing N2A cells (Fig. 3 C). mESCs (2 × 10⁵) were then incubated in 2 ml N2B27 medium for 24 h with EVs purified from control or GFP-expressing cells. mESCs were then harvested by centrifugation and washed with PBS twice, and the GFP signal was detected in the lysate of the receiving ESCs (Fig. 3 D). In incubations of mESCs in a wide range of EV concentrations, GFP was detected associated with cells at GFP-EV levels of ≥2 × 10⁹ (Fig. 3 E). In a time course, we observed GFP-EV uptake within 3 h and progression with continued incubation (Fig. 3 F). We next compared the uptake of free GFP (10 ng) to GFP-EVs (~8 × 10⁹ EVs contain ~10 ng GFP) added in equivalent amounts, as detected by quantitative immunoblot, to mESCs incubated in 2 ml N2B27 medium. GFP-EVs were more rapidly and efficiently internalized (Fig. 3 G). These results indicate that mESCs internalize lumenal soluble as well as membrane constituents of EVs in a time- and a dosage-dependent manner.

**Buoyant density–purified EVs from differentiated neuronal cells promote mESC neural induction**

mESCs default to a neural progenitor fate in serum-free growth medium (Ying et al., 2003). To explore the effects of differentiated neuronal EVs during this process, mESCs were treated

with purified EVs in serum-free medium for 6 d. The medium containing EVs was changed every day, with fresh EVs added each day. Cells were harvested for gene expression analysis by qPCR. We found that the neural stem cell marker *nestin*, and a neuronal marker gene *Six3*, were up-regulated twofold in comparison to an EV-free control by EVs from NGF-induced, but not by EVs from uninduced, PC12 cells (Fig. 4 A). EVs derived from PC12 cells that had been induced by NGF for 6 d (N6-EVs) caused the most robust up-regulation of neural markers at mESC differentiation day 4 (Fig. 4, A and C). This time point was used in subsequent experiments. Neural marker up-regulation by N6-EVs was dose dependent (Fig. 4 B). Furthermore, two other neural progenitor genes, *Pax6* and *Sox1*, and neuronal marker *Tuj1*, were all up-regulated by N6-EVs (Fig. 4 C). Although NGF is known to stimulate neural gene expression in mESCs, we found that addition of a neutralizing NGF antibody to an incubation of N6-EVs with mESCs did not affect the level of expression of these neural markers, suggesting that possible residual NGF in the EV preparation did not account for this effect (Fig. 4 C). The neutralizing activity of NGF antibody to NGF was confirmed by cell morphology and marker gene expression analysis (Fig. S3, A and B). Immunostaining confirmed the increase of expression of Nestin and Pax6 in neural progenitor cells after treatment of mESCs with EVs from NGF-treated PC12 cells (Fig. 4, D and E). EVs derived from 12-d differentiated neuronal ESCs (ES D12-EV) up-regulated neural markers at mESC differentiation day 4, whereas EVs secreted from 8 d EBs (ES D8-EV), or the EVs released from pluripotent mESCs (ES D0-EV; Fig. 4 F), were much less active.

Similar results were obtained with EVs isolated from RA-induced N2A cells (Fig. S3, C and D). Immunostaining showed an increase of Nestin and Pax6 expression in neural progenitor cells after treatment with RA-induced EVs (Fig. S3, E and F). These results suggest that EVs from neuronal cells promote a neural fate commitment of mESCs.

During the mESC serum-free neural differentiation process, we found that EVs from NGF-induced PC12 cells appeared to increase the size and density of cell clusters (Fig. 4 G). Likewise, an increase in mESC cell number from ~5.2 × 10⁴ to ~7.5 × 10⁴/ cm² was observed after 4 d of treatment with EVs from NGF-induced PC12 cells (Fig. 4 H). This result was affirmed using a BrdU cell proliferation assay (Fig. 4 I).

**Cyclin D1 is sorted into EVs**

We considered the possibility that EVs from differentiated PC12 and N2A cells may transfer proteins that could influence the fate and proliferation of stem cells. One candidate, cyclin D1, a cell cycle regulator, was reported to promote neural fate conversion in human ESCs (Pauklin and Vallier, 2013; Pauklin et al., 2016). We examined the expression and sorting of the three paralogs of

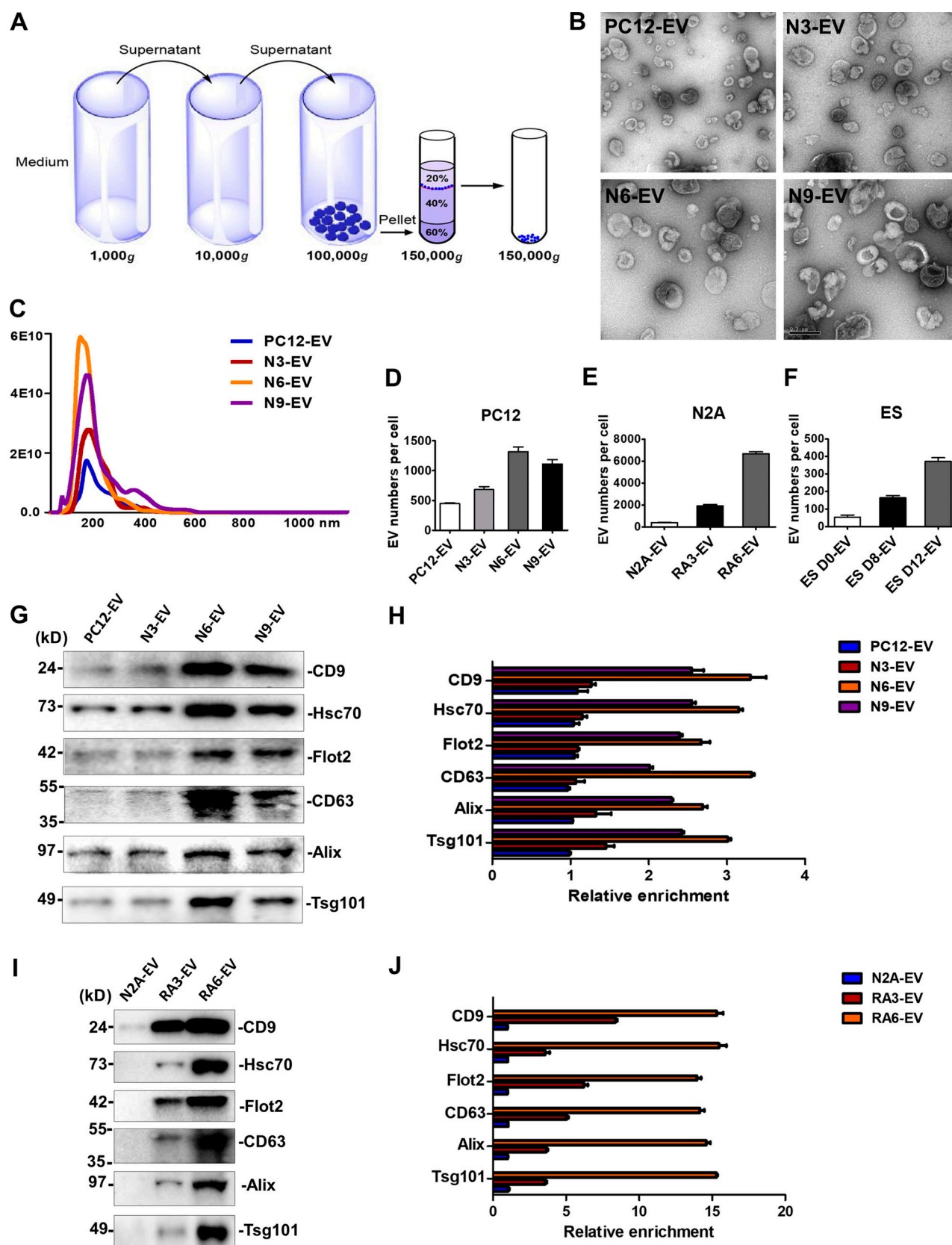

Figure 2. **EV production increased during neuronal differentiation. (A)** Schematic of the EV purification strategy. **(B)** Representative electron micrographs of negatively stained samples of purified EVs at 9,300× magnification. Purified EVs from untreated PC12 cells cultured for 3 d (PC12-EV) or treated with NGF for 3, 6, and 9 d (N3-EV, N6-EV, and N9-EV). During PC12 differentiation, EVs were collected from 3-d-cultured cells, and fresh medium together with NGF were replaced every 3 d. Scale bar, 0.2 μm. **(C)** Nanoparticle tracking analysis of the size distribution and the number of purified EVs from 420-ml medium of

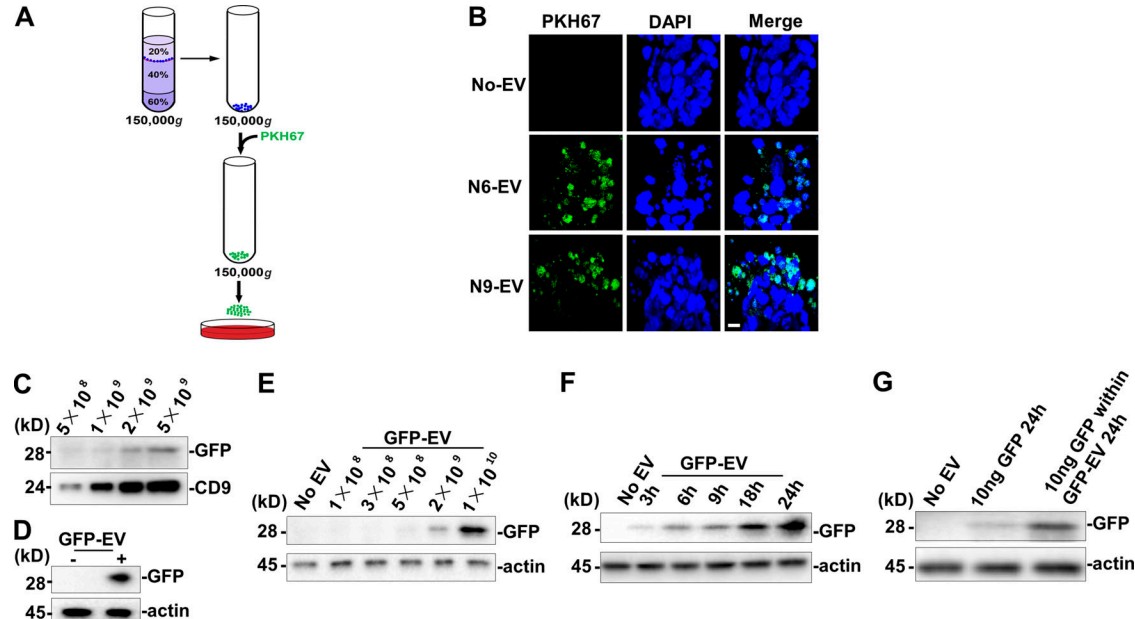

untreated PC12 cells cultured for 3 d or treated with NGF for 3, 6, and 9 d. **(D)** The number of EVs released per PC12 cell untreated or treated with NGF for 3, 6, and 9 d. EV number was quantified by nanoparticle tracking analysis. Cell number was quantified with a hemocytometer. The values represent the mean ± SD, from three independent experiments. Error bars represent SD from independent samples. **(E)** The number of EVs released per N2A cell cultured for 3 d or treated with RA for 3 and 6 d. During N2A differentiation, EVs were collected from 3-d-cultured cells, and fresh medium together with RA were replaced every 3 d. The values represent the mean ± SD, from three independent experiments. Error bars represent SD from independent samples. **(F)** The number of EVs released per ESC untreated or differentiated for 8 and 12 d. During ES differentiation, EVs were collected from 2-d-cultured cells, and fresh medium was replaced every 2 d. The values represent the mean ± SD, from two independent experiments. Error bars represent SD from independent samples. **(G)** Immunoblots of CD9, Hsc70, Flot2, CD63, Alix, and Tsg101 in EVs from the same number of cells. $2 \times 10^7$ PC12 cells were untreated or treated with NGF for 3, 6, and 9 d. **(H)** Quantitative analysis of the immunoblots in G. The values represent the mean ± SD, from two independent experiments. Error bars represent SD from independent samples. The signal in PC12-EV group was set as 1. **(I)** Immunoblots of CD9, Hsc70, Flot2, CD63, Alix, and Tsg101 in EVs from the same number of cells. $2 \times 10^7$ N2A cells were untreated or treated with RA for 3 and 6 d. **(J)** Quantitative analysis of the immunoblots in I. The values represent the mean ± SD, from two independent experiments. Error bars represent SD from independent samples. The signal in N2A-EV group was set as 1.

---

cyclin D (D1, D2, and D3) in differentiated PC12 cells and EVs. Cyclins D1–D3 were gradually up-regulated from day 0 to 9 during NGF treatment (Fig. 5 A). Only cyclins D1 and D2, but not D3, were detected in the EVs from differentiating cells (Fig. 5 B). Compared with the moderate increase of cyclin D proteins detected in cell lysates, a greater enrichment of cyclin D1, and to a lesser extent D2, in EVs was detected at day 6 (Fig. 5, B and C). Specific cyclin D1 sorting was also enriched in EVs from RA-induced N2A cells, especially at day 6 (Fig. S4 A). Compared with EVs from nondifferentiated mESCs and EVs from 8-d differentiated EBs, significant cyclin D1 was enriched in EVs from 12-d differentiated neurons (Fig. 5 D). The cyclin D–dependent kinase family member CDK4 was observed in EVs (Figs. 5 E and S4 A). Flot2, which was used as a positive EV membrane protein control, was present in both EVs and cells, whereas the cis-Golgi matrix protein, GM130, was detected only within the cells (Fig. 5

E). Given that cyclin Ds drive the G1/S phase transition, we tested other cyclins that contribute to cell-cycle progression at S phase and M phase and found that cyclin B1, cyclin E1, and cyclin A2 were expressed in cells but not in EVs (Fig. 5 E). Likewise, other cell cycle–related factors, including pRB, p57, p27, p21, and pErk, were detected in cells but not in EVs (Fig. S4 B). These results suggest that at least some factors specific to the G1/S phase transition were sorted into EVs, notably cyclin D1.

We used a standard protease protection assay to probe the localization of cyclin D1 with respect to the EV membrane. The EV marker proteins, Hsc70, Flot2, Tsg101, and Alix, were dramatically sensitive to proteolytic degradation by 10 µg/ml proteinase K in the presence of Triton X-100, but not in the absence of detergent (Fig. 5 F). We conclude that these proteins are enclosed within the interior of the EV. Similarly, cyclin D1, cyclin D2, and CDK4 were all sensitive to proteinase K but protected in

**Figure 3. Differentiated neuronal EVs taken up by mESCs. (A)** Schematic of mESCs treated by PKH6 dye–labeled EVs. **(B)** Confocal images of differentiated mESCs incubated without EV (No-EV) or with PKH6 dye–labeled N6 or N9 EV. Nuclei were stained with DAPI. Scale bar, 10 µm. **(C)** Immunoblot of GFP and CD9 in EVs (GFP-EV) purified from GFP-expressing cells. $2 \times 10^5$ mESCs in a 35-mm dish were incubated in 2 ml of N2B27 medium for 24 h with EVs purified from control or GFP-expressing cells. **(D)** Immunoblot of GFP from whole-cell lysate of mESCs treated with PBS or GFP-EV for 24 h. **(E)** Immunoblot of GFP from whole-cell lysate of mESCs treated with indicated doses of GFP-EV for 24 h. **(F)** Immunoblot of GFP from whole-cell lysate of mESCs treated with indicated time of incubation with GFP-EV. **(G)** Immunoblot of GFP from whole-cell lysate of mESCs treated with 10 ng GFP protein for indicated time or treated with GFP-EV containing 10 ng GFP for 24 h. The GFP protein amount within the GFP-EV was detected by quantitative immunoblot.

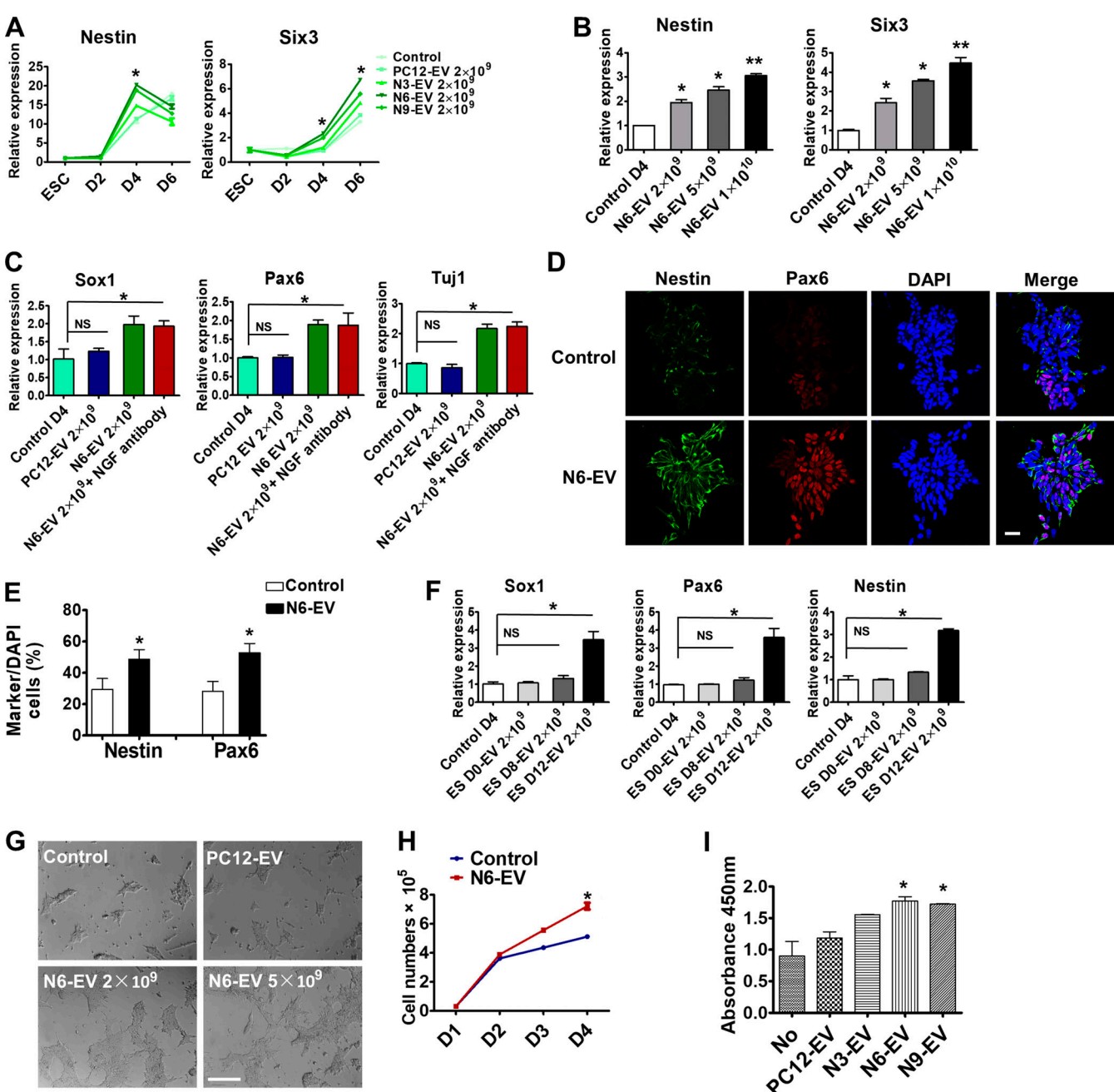

Figure 4. **Neuronal EVs promote neural induction and cell proliferation of mESCs. (A)** Gene expression analysis of *Nestin* and *Six3* in mESCs treated without (control) or with indicated EVs. EV number was quantified by Nanosight 2000. Data plotted were from three independent experiments, each with triplicate qPCR reactions; error bars represent SD from independent samples. The values represent the mean ± SD (*, P < 0.05). **(B)** Gene expression analysis of *Nestin* and *Six3* in mESCs treated for 4 d without (control) or with different doses of N6-EV. Data plotted were from three independent experiments, each with triplicate qPCR reactions; error bars represent SD from independent samples. The values represent the mean ± SD, from three independent experiments (*, P < 0.05; **, P < 0.01). **(C)** Gene expression analysis of *Sox1*, *Pax6*, and *Tuj1* in mESCs treated for 4 d with 2× 10⁹ PC12-EV, N6-EV, or N6-EV together with NGF neutralizing antibody (500 ng/ml). The values represent the mean ± SD, from three independent experiments (*, P < 0.05; NS, P > 0.05). Error bars represent SD from independent samples. **(D)** Immunostaining of Nestin (green, Alexa Fluor 488) and Pax6 (red, Alexa Fluor 568) in mESCs as described in B. Scale bar, 25 μm. **(E)** Quantitative analysis of the percentage of cells containing indicated markers compared with DAPI-stained cells. The values represent the mean ± SD, from three independent experiments (*, P < 0.05). Error bars represent SD from independent samples. **(F)** Gene expression analysis of *Sox1*, *Pax6*, and *Nestin* in mESCs treated for 4 d with 2 × 10⁹ EVs from undifferentiated pluripotent ESCs (ES-D0), EBs of ES differentiated for 8 d in KSR medium (ES-D8), and EBs trypsinized in N2 medium for an additional 4 d (ES-D12). The values represent the mean ± SD, from two independent experiments (*, P < 0.05; NS, P > 0.05). Error bars represent SD from independent samples. **(G)** The cellular morphology of mESCs treated with PC12-Exo or different doses of N6-EV in N2B27 medium for 4 d. Scale bars, 200 μm. **(H)** Quantitative analysis of mESC number treated with or without N6-EV. Data plotted were from two independent experiments. The values represent the mean ± SD (*, P < 0.05). Error bars represent SD from independent samples. **(I)** Proliferation analysis of mESCs with BrdU staining after EV treatment. The values represent the mean ± SD, from two independent experiments (*, P < 0.05). Error bars represent SD from independent samples.

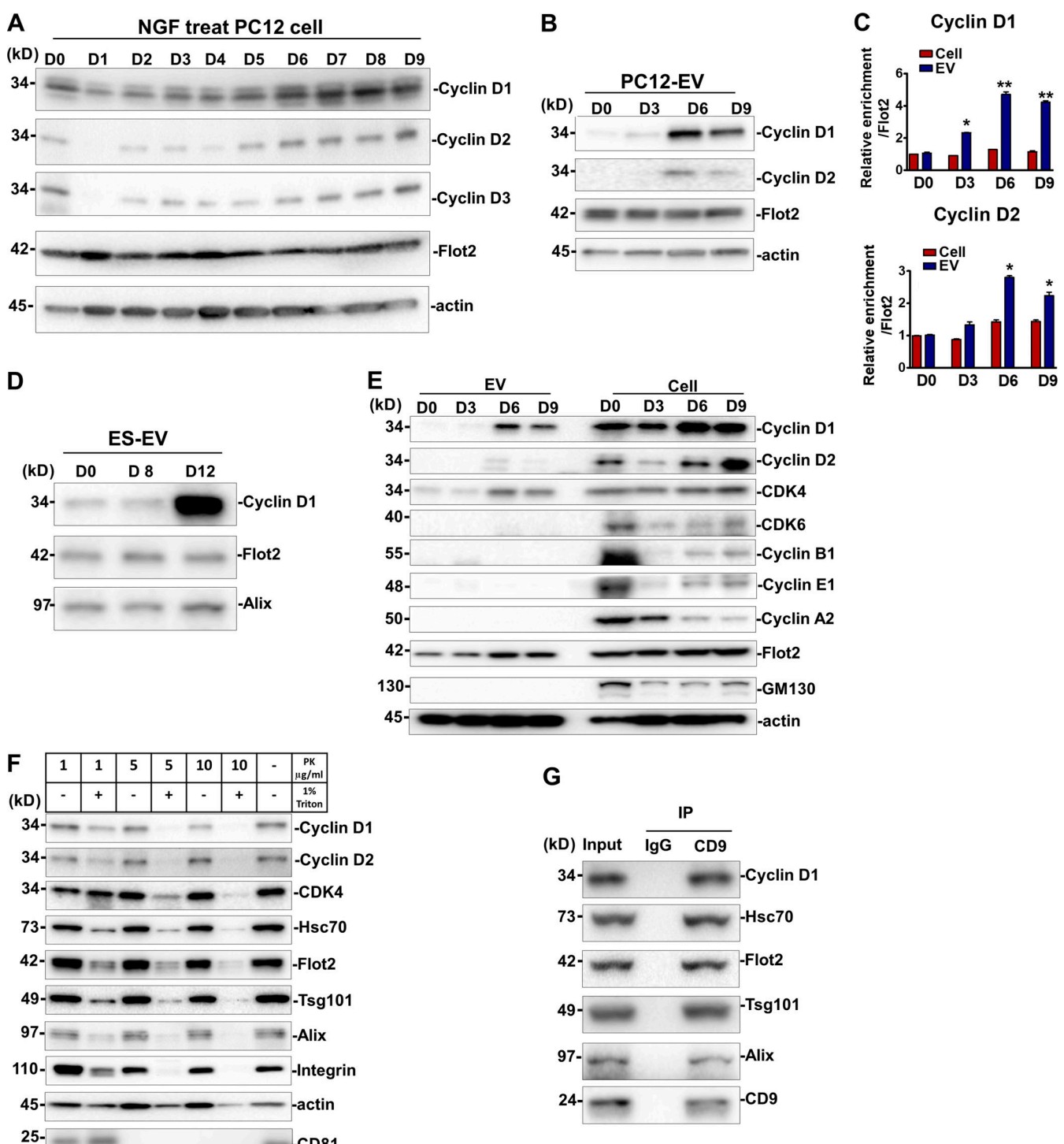

Figure 5.  **Cyclin D1 enriched in EVs during neurogenesis. (A)** Immunoblot analysis of cyclin D1, 2, and 3 in PC12 cells induced by NGF for different times. D0, PC12 cells without NGF treatment. D1–D9, PC12 cells incubated with NGF for 1–9 d. **(B)** Immunoblot analysis of cyclin D1/2 in EVs purified from PC12 cells (D0) and EVs purified from NGF-induced PC12 cells for 3, 6, and 9 d (D3, D6, and D9). **(C)** Quantitative immunoblot analysis of protein levels described in B. The D0 signal was set as 1. Flot2 signal was used as a internal control. The values represent the mean ± SD, from three independent experiments (*, P < 0.05; **, P < 0.01). Error bars represent SD from independent samples. **(D)** Immunoblots of cyclin D1, Flot2, and Alix in EVs from undifferentiated ESCs (ES D0-EV) or 8-d (ES D8-EV) or 12-d (ES D12-EV) differentiated ESCs. **(E)** Immunoblots of cyclins, CDKs, Flot2, GM130, and actin in EVs and whole-cell lysates of PC12 cells or NGF-induced PC12 cells. **(F)** Immunoblot analysis of cyclin D1/2 and multiple EV markers of N6-EVs treated with different concentrations of proteinase K (PK), with or without 1% Triton X-100. **(G)** Immunoblots for cycinD1, Alix, Hsc70, Tsg101, and CD9 after immunoprecipitation of 5 × 10^10 N6-EV with anti-CD9 antibody. IP, immunoprecipitates.

the absence of Triton X-100 (Figs. 5 F and S4 C). Tetraspanin membrane protein, CD81, was subjected to proteolytic degradation by 5 µg/ml proteinase K (Fig. 5 F). We then used an antibody directed to membrane-exposed epitopes of the tetraspanin membrane protein, CD9, to immunoisolate intact vesicles and probe the coincident localization of cyclin D1 in EVs. An immobilized form of CD9 antibody coimmunoprecipitated cyclin D1 along with EV markers Hsc70, Tsg101, and Alix (Fig. 5 G). These results support the conclusion that cyclin D, especially cyclin D1, is sorted into the luminal interior of EVs produced by differentiating PC12 and N2A cells.

## Hsc70 facilitates cyclin D1 package into EVs

We used ascorbic acid peroxidase (APEX) proximity labeling to detect proteins in contact with cyclin D1 during RA-induced differentiation of N2A cells (Hung et al., 2016). N2A cells stably expressing cyclin D1–APEX were obtained by lentivirus-mediated gene delivery. Biotin-phenol (B) and $H_2O_2$ (H) were added to cells separately or in combination (B+H). Equal amounts of protein, as estimated by Ponceau S staining of gels, were evaluated in samples from three incubations (Fig. 6 A). Streptavidin-HRP was used to label biotinylated proteins, which were detected primarily in the sample incubated with B+H (Fig. 6 A). The biotinylated proteins were precipitated with streptavidin beads, and bound proteins were analyzed by mass spectrometry (MS). We identified proteins that were enriched in the B+H group compared with the B group, with an increase in at least280 unique proteins (Fig. 6 B and Data S1). 312 unique peptides of the heat shock protein, Hsp90b1, were identified in the B+H group, with only nine peptides in the B group. Other unique differences were seen among several Hsps (Fig. 6 B). We thus chose Hsp90 and Hsc70 to examine direct interactions with cyclin D1 and CDK4, the two proteins that we found to be selectively sorted into EVs during RA-induced differentiation of N2A cells.

In the immunoprecipitation experiments, we found that Hsc70-HA but not Hsp90-HA coprecipitated cyclin D1–Flag and CDK4-Flag in N2A cells (Fig. 6 C). The converse experiment, with anti-Flag antibody, was also performed in PC12 cells, where we found that cyclin D1–Flag coprecipitated with Hsc70-HA (Fig. 6 D). Similarly, EVs ($5 \times 10^{10}$) from RA-induced N2A cells contained cyclin D that coimmunoprecipitated with Hsc70 (Fig. 6 E).

We then tested the possibility that Hsc70 function may be important in the sorting of cyclin D1 into EVs. VER-155008 (VER) is a potent and selective inhibitor of Hsc70 ATPase activity that has been used to assess Hsc70 function in cellular processes (Massey et al., 2010). We purified EVs from 4-d differentiated N2A cells treated with or without 5 µM VER for another 2 d. EVs isolated from treated or untreated cells displayed roughly equivalent levels of Alix and CD9, indicating that VER treatment did not change the overall number of EVs being secreted. Consistent results were obtained by particle tracking assay. In contrast, cyclin D1 levels in EVs declined 2.2-fold during the period of treatment with VER (Fig. 6 F), suggesting a potential role for Hsc70 in cyclin D1 packaging into EVs.

We then tested whether a nonfunctional form of Hsc70 would alter the loading of EVs with cyclin D1. Site-directed mutation of

Asn to Asp-10 (D10N) abolishes the ATPase activity of Hsc70 (Huang et al., 1993). Correspondingly, the quantity of cyclin D1 in EVs was decreased in cells transiently expressing the Hsc70 D10N mutant compared with WT Hsc70 (Fig. 6 G).

Finally, we examined whether EVs collected from differentiated N2A cells pretreated with or without VER had different effects on gene expression in mESCs. EVs ($2 \times 10^9$) were collected and incubated with mESCs in serum-free medium for 4 d. The neural progenitor–specific genes, *Pax6* and *Nestin*, together with the neuronal marker, *Six3*, were up-regulated by the EVs from control cells but not EVs from VER-pretreated cells (Fig. 6 H). In addition, we used lentivirus-mediated CRISPRi delivery to knock down Hsc70 in N2A cells (Gilbert et al., 2014). A control of RA-EVs collected from dCas9 cells, but not the EVs from Hsc70 sgRNA transfected cells, up-regulated the expression of *Pax6* and *Nestin* (Fig. 6 J). These results suggested that the heat shock protein Hsc70 contributes to cyclin D1 sorting and may serve a direct or indirect role in promoting the function of EVs in the differentiation of mESCs.

## Cyclin D1 is required for EV-mediated neural induction of mESCs

To determine whether EV cyclin D1 was taken up by mESCs, cyclin D1–GFP EVs were incubated with mESCs in N2B27 medium. After 24 h, green puncta were detected overlapping the endogenous CD9-labeled mESC cytoplasm, but rarely in the nucleus (Fig. 7 A). After 4 d of daily addition of cyclin D1–GFP EVs to mESCs, ~30% of total cyclin D1 was detected by immunoblot of cell lysates at the position of migration of the hybrid protein. The level of endogenous cyclin D1 also increased during the 4-d incubation (Fig. 7 B). These results suggested that EV cyclin D1 was internalized by mESCs in serum-free conditions of growth.

Fluorescence localization of the internalized cyclin D1–GFP offered inadequate spatial resolution to determine if the content of EVs was delivered to the cytoplasm of mESCs. As an alternative approach, we examined the proteins in contact with internalized cyclin D1–APEX in comparison to those found in isolated EVs. EVs were collected from cyclin D1–APEX-expressing N2A cells (Fig. 7 C). In incubation with isolated EVs, the B+H-treated EVs revealed multiple biotinylated proteins that were not detected in incubations containing biotin-phenol without $H_2O_2$ (Fig. 7 D). Cyclin D1–APEX EVs were then incubated with mESCs for 2 d in serum-free medium (Fig. 7 E). Subsequently, these cells were incubated with APEX reagents, which resulted in the appearance of biotinylated proteins in a range from ~25 to ~130 kD in the B+H group (Fig. 7 F). MS of mESC cell proteins from a streptavidin pull-down assay identified 116 proteins enriched in the B+H treatment group, with only 3 proteins detected in the biotin-phenol control group (Fig. 7 G). Gene ontology (GO) analysis indicated that specific enriched proteins were most related to mRNA and ribonucleoprotein binding partners (Fig. 7 H).

To distinguish the possibility that the biotinylated proteins were from cyclin D1–APEX in contact with mESC proteins as opposed to those in the EV donor vesicle, we analyzed two candidates, the primarily nuclear-localized proteins Lin28 and

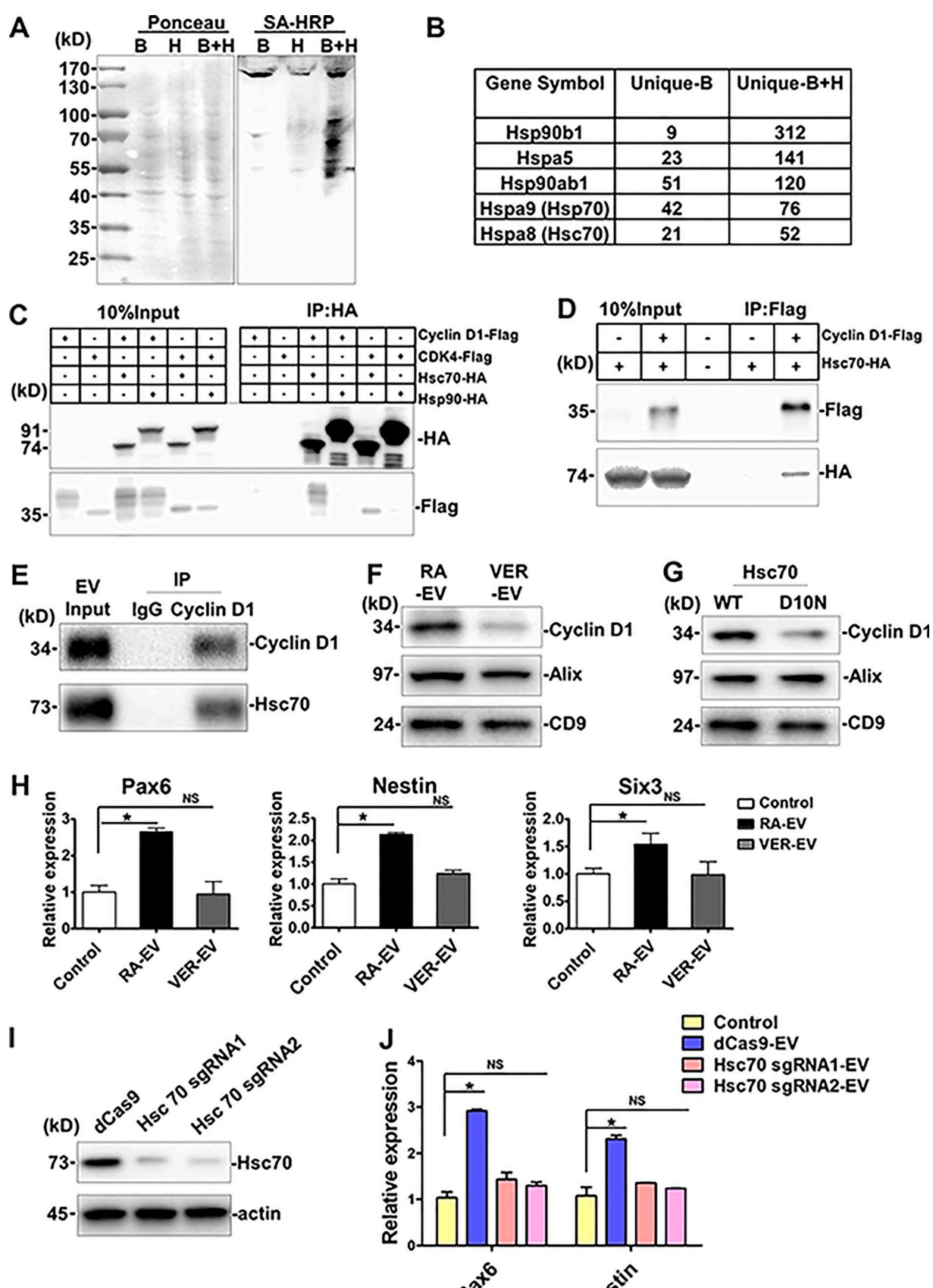

Figure 6. **The chaperone protein Hsc70 facilities cyclin D1 package into EVs. (A)** Characterization of APEX-mediated proximity biotinylation of cyclin D1 protein targets by blotting with streptavidin. Cyclin D1–APEX fusion gene was delivered into N2A cells by lentivirus infection. Biotinylated protein was detected

by blotting with streptavidin (SA)-HRP. Ponceau S staining (left) of the same membrane served as loading control. **(B)** Table showing MS analysis of the unique peptides in biotin-phenol together with $H_2O_2$ (B+H) or without $H_2O_2$ (B). **(C)** CoIP analysis of Hsc70 and Hsc90 with cyclin D1 and CDK4 in N2A cells. **(D)** CoIP of cyclin D1 and Hsc70 in PC12 cells. **(E)** CoIP of cyclin D1 and Hsc70 in $5 \times 10^{10}$ RA-EVs. **(F)** Immunoblots of cyclin D1, Alix, and CD9 in EVs collected from the differentiated N2A cells treated with VER-155008 (VER). N2A cells pretreated with RA-containing differentiation medium for 4 d, after which cells were exposed to fresh differentiation medium with or without 5 µM VER for two more days. EVs collected from 6-d differentiation of N2A cells. **(G)** Immunoblots of cyclin D1, Alix, and CD9 in EVs collected from the differentiated N2A cells transfected with WT Hsc70 (WT) or D10N mutant Hsc70 (D10N; >50% transfection efficiency). WT Hsc70 or D10N mutant Hsc70 were transfected by Lipofectamine 2000 in seven plates of 70%-confluency N2A cells in DMEM medium for 10 h, followed by a change to fresh differentiation medium for 3 d. EVs were collected from both cells. **(H)** Expression analysis of *Pax6*, *Nestin*, and *Six3* in differentiated mESCs treated with $2 \times 10^9$ EVs from RA-induced N2A cells with (VER-EV) or without (RA-EV) VER. EVs were collected as described in F. The values represent the mean ± SD, from three independent experiments (*, P < 0.05; NS, P > 0.05). Error bars represent SD from independent samples. **(I)** Immunoblots of Hsc70 and actin in control or Hsc70 sgRNA–transfected N2A cells. dCas9 was stably expressed in N2A cells by lentivirus (dCas9), Lentivirus was then used to introduce Hsc70 sgRNA1/2 by transfection of dCas9 cells. **(J)** Expression analysis of *Pax6* and *Nestin* in differentiated mESCs treated with $2 \times 10^9$ EVs from RA-induced N2A dCas9 cells or Hsc70 sgRNA–transfected cells. Values represent the mean ± SD, from three independent experiments (*, P < 0.05; NS, P > 0.05). Error bars represent SD from independent samples.

nucleolin, found in the MS analysis described above. Neither of these proteins was detected in the sample of isolated EVs (Fig. 7 I). However, each was detected in a biotinylated form in the cells exposed for 2 d to cyclin D1–APEX EVs (Fig. 7 I). These results indicated that the cyclin D1 content of EVs isolated from RA-induced N2A cells is productively taken up and delivered to the cytoplasm or nucleoplasm of mESCs.

Next, to examine the contribution of EV cyclin D1 to mESC neural commitment, we generated cyclin D1–overexpressing N2A cells by use of a lentiviral vector. The empty vector without cyclin D1 overexpression was used as a control. Overexpression led to 2.1-fold more cyclin D1 protein packaged into EVs (Fig. 7 J). We found that treatment of mESCs with high cyclin D1 EVs increased the expression of neural marker genes *Pax6* and *Six3* (Fig. 7 K), as well as of Pax6 expressed in neural progenitor cells (Fig. 7 L). To explore if EV cyclin D1 was necessary to promote neural differentiation, we generated a cyclin D1 knockout N2A cell line using CRISPR/Cas9. The EVs purified from the knockout cells had similar levels of Tsg101 but lacked cyclin D1 (Fig. 7 M). The whole protein profile did not show significant changes after cyclin D1 knockout (Fig. S5 A). The expression of neural marker genes *Pax6* and *Six3* was significantly lower in cells treated with cyclin D1–depleted EVs than with EVs from control RA-treated N2A cells (Fig. 7 N). Pax6-positive cells were reduced in the cyclin D1 KO EV treatment group versus the samples incubated with a control EVs from RA-treated N2A cells (Fig. 7 P). Taken together, these results indicated that EV cyclin D1 accelerated the induction of neural fate in mESCs.

## Discussion

In this study, we report substantial changes in the abundance, physical and functional properties, and content of EVs produced during neuronal differentiation of neural cell lines PC12 and N2A. Cyclin D1 and CDK4 were selectively sorted into EVs during differentiation mediated by NGF (for PC12 cells) and RA (for N2A cells). EVs from differentiating cells fractionated by rate and buoyant density centrifugation promoted the expression of genes characteristic of neural induction in mESCs. Cyclin D1 was of particular importance in stimulating this differentiation, and Hsc70, a constitutive component of EVs, played a role in the EV capture of cyclin D1 (Fig. 8).

### The secretion of EVs in neuronal cells

Previous studies have shown that EVs are released at different stages of neural cell development (Janas et al., 2016). Of particular importance are examples where cellular communication mediated by EVs produced by neural stem cells has been suggested to mediate cytokine signaling in target cells (Cossetti et al., 2014). Other studies have addressed the regulation of EV production, for example in the controlled release of EVs by primary undifferentiated cortical neurons (Fauré et al., 2006) and in a role for calcium influx in fully differentiated cortical and hippocampal neurons, where glutamatergic synaptic activity promoted EV secretion (Lachenal et al., 2011). In more recent studies, the neuronal activity–dependent secretion of Arc protein enclosed within extracellular particles has been proposed to promote the intercellular transfer of RNA (Ashley et al., 2018; Pastuzyn et al., 2018). Little is known about the changes in EV content and function as they relate to the differentiation of neuronal precursor cells.

EVs derive from the cell surface by a process of membrane budding to produce particles that are termed microvesicles. A distinct population of EVs, termed exosomes, arise by invagination of membrane into endosomes where intralumenal vesicles accumulate in a structure called the generate MVB. MVBs have two possible fates: fusion with the lysosome, which results in the degradation of the intralumenal vesicles, or fusion with the cell surface membrane, which results in the secretion of the intralumenal vesicles. In previous work from our laboratory, we showed that EVs can be broadly separated into two populations on an iodixanol density gradient, with microvesicles sedimenting to a lower and exosomal EVs to a higher buoyant density (Temoche-Diaz et al., 2019).

In neurons, MVBs are differentially distributed between divergent neuronal compartments, including cell bodies and dendrites (Von Bartheld and Altick, 2011). Here we used NGF-induced PC12 cells, RA-induced N2A cells, and EBs derived from mESCs differentiated into Tuj1⁺ neurons as simple models that recapitulate neurite extension in vitro (Greene and Tischler, 1976; Tremblay et al., 2010; Bibel et al., 2007). Using a two-step fractionation procedure, we found dramatic increases in EV production and in the buoyant density profile of EVs produced during the differentiation of each cell type. Consistently, our results showed that the EV marker proteins CD9 and Hsc70 became more heterogeneous in terms of membrane buoyant

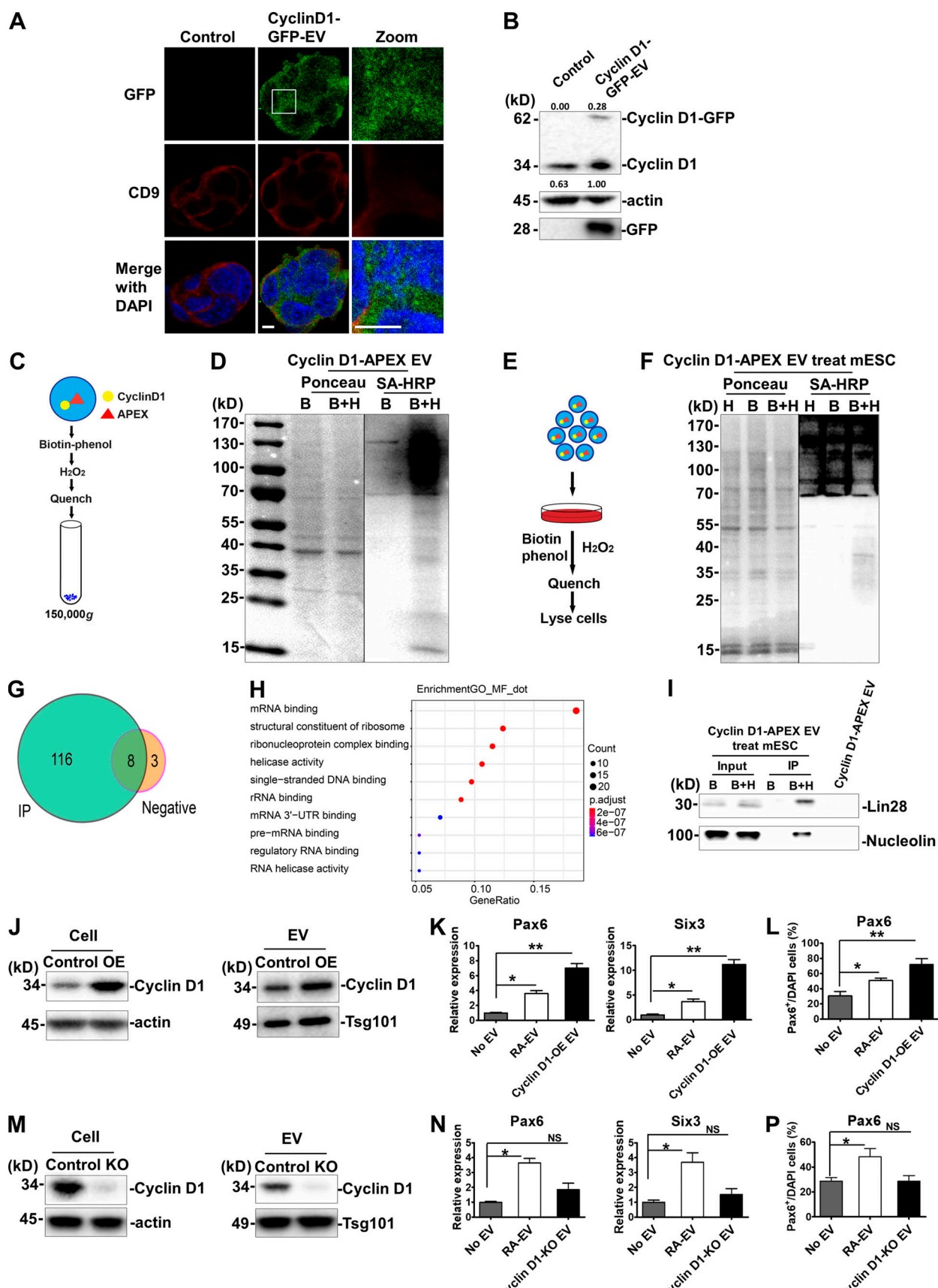

Figure 7. **Cyclin D1 is important for EV-mediated neural induction of mESCs. (A)** Immunostaining of GFP (green, Alexa fluor 488) and CD9 (red, Alexa fluor 568) in differentiated mESC cells without (control) or with cyclin D1–GFP EV treatment. Magnified view is shown in panel 3. Nuclei were stained with DAPI.

Scale bars, 5 µm. **(B)** Immunoblots of cyclin D1, actin, and GFP of differentiated mESCs without incubation or incubated for 4 d with cyclin D1–GFP EVs. Quantification of fusion protein uptake was calculated as the ratio of exogenous cyclin D1–GFP to endogenous cyclin D1. **(C)** Schematic of biotinylation labeling of cyclin D1–APEX EVs. **(D)** Streptavidin-HRP blotting analysis of biotinylated proteins in cyclin D1–APEX-expressing EVs. EVs were treated with biotin-phenol together with $H_2O_2$ (B+H) or not (B). Biotinylated protein was detected by blotting with streptavidin (SA)-HRP. Ponceau S staining (left of panel) of the same membrane serves as loading control. **(E)** Schematic of mESCs treated with cyclin D1–APEX EVs and biotinylated proteins labeled in differentiated mESCs. **(F)** SA-HRP blotting of biotinylated proteins in mESCs treated with cyclin D1–APEX EVs. **(G)** Venn diagram of the MS results. MS sample was collected as described in Materials and methods. Immunoprecipitation with streptavidin was used to enrich the biotinylated proteins. Diagram generated by Venn diagram package in the R program for statistical computing. **(H)** GO analysis of the MS results shown in G. GO analysis was generated by topGO package in the R program for statistical computing. **(I)** After the treatment described in E and F, immunoblots of Lin28 and nucleolin in differentiated mESCs treated with cyclin D1–APEX EVs. **(J)** Cyclin D1 was increased in the EVs from N2A cells overexpressing cyclin D1 (OE). The protein level of cyclin D1 was detected in control and OE samples. Actin was used as the internal control of whole-cell lysate, and Tsg101 was used as the loading control of EVs. **(K)** Gene expression level of *Pax6*, *Six3*, and *Map2* was determined in differentiated mESCs treated without EVs and with RA-EV or OE EVs. The values represent the mean ± SD, from three independent experiments (*, P < 0.05; **, P < 0.01). Error bars represent SD from independent samples. **(L)** Quantitative analysis of the percentage of cells containing Pax6 normalized to DAPI stain in differentiated mESCs treated without EVs and with RA-EV or OE EVs. The values represent the mean ± SD, from two independent experiments (*, P < 0.05). Error bars represent SD from independent samples. **(M)** Cyclin D1 was absent from cyclin D1 knockout N2A cells and the EVs from cyclin D1 knockout (KO) N2A cells. The cyclin D1 protein was detected in control and KO samples. **(N)** The expression of *Pax6*, *Six3*, and *Map2* was analyzed in differentiated mESCs treated without EVs and with RA-EV or cyclin D1 KO EVs. The values represent the mean ± SD from three independent experiments (*, P < 0.05; NS, P > 0.05). Error bars represent SD from independent samples. **(P)** Quantitative analysis of the percentage of cells containing Pax6 normalized to DAPI stain in differentiated mESCs treated without EVs and with RA-EV or cyclin D1 KO EVs. The values represent the mean ± SD, from two independent experiments (*, P < 0.05). Error bars represent SD from independent samples.

density during differentiation in vitro (Fig. 1, D and E). These changes make it difficult to classify the EVs as either microvesicles or exosomal EVs. Whichever biogenesis path is used to produce the EVs from differentiating PC12 or N2A cells, their content of selected soluble cell cycle regulatory proteins and change in functional characteristics suggest an active role for cargo sorting in their production (Figs. 5 and S5).

### Sorting of cyclin D1/CDK4 into EVs
Neurons are believed to have lost their capacity to proliferate once they are terminally differentiated (Frade and Ovejero-Benito, 2015; Ohnuma and Harris, 2003). Cyclin D is synthesized at the beginning of G1, and it binds and activates CDK4/6 when the cell leaves the quiescent state (van den Heuvel, 2005;

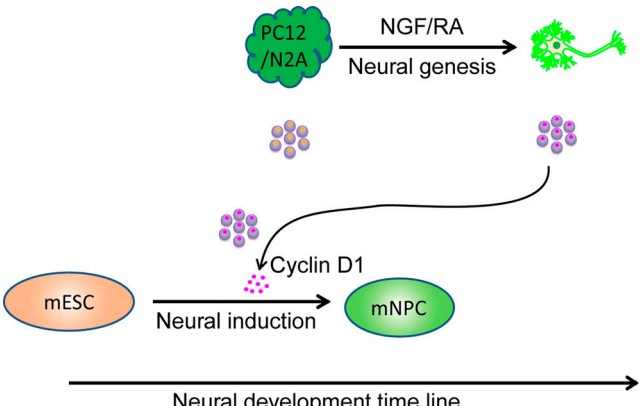

Figure 8. **Model.** Neural development includes early-stage neural induction and late-stage neural genesis. During neural genesis, PC12 or N2A cells (dark green) respond to NGF or RA to differentiate into neuronal cells (bright green). The content of EVs exhibits dynamic changes corresponding to the fate conversion. Cyclin D1 (magenta dots inside the purple EVs) was enriched in EVs from differentiating neurons. Additional cyclin D1 enriched in EVs from the neuronal cells accelerates the commitment of mESCs (light orange) to neural progenitor cells (mNPC, light green). Exosomal communication between different development stages may contribute to commitment and conversion of mESCs to the neural lineage.

Dehay and Kennedy, 2007). Here, we found cyclin D1 and CDK4 enriched in purified EVs from three different model sources of neuronal cells. In contrast, the regulators of G2/M phase transition, cyclin A2 and cyclin E1, were detected in differentiated cells but not in EVs (Fig. 5 D). This sorting fidelity was reproduced during RA-induced differentiation of the neuroblastoma cell line N2A (Fig. S5, A and C). Of possible relevance to our findings, many regulators of the G1/S transition are detected in the adult mouse brain. These may be deployed for cell cycle reentry under pathological conditions such as in response to DNA damage and oxidative stress (Klein and Ackerman, 2003). EV-mediated secretion may be a protective response to reduce the possibility of abortive cell cycle reentry.

The cyclin D family may also play important roles in neural development (Lukaszewicz and Anderson, 2011). In the mouse cortex and hippocampus, overexpression of cyclin D1/CDK4 delays neurogenesis and promotes expansion of basal neural progenitors by shortening the G1 phase (Lange et al., 2009; Pilaz et al., 2009). This may relate to our observation of an increase in the expression of cyclin D family members during NGF-induced differentiation (Fig. 5 A), consistent with other reports. The secretion of EVs enriched in cyclin D1 and their uptake in stem cells may reinforce this aspect of neuronal differentiation. Our findings on EV-mediated cyclin D1 secretion and transfer in a neuronal lineage may extend to astrocytes and oligodendrocyte lineages, which have been shown previously to express cyclin D1 (Nobs et al., 2014; Ma et al., 2015; Bosone et al., 2001). Of course, other pathways almost certainly play a role in the differentiation of neural progenitor cells. Parthasarathy et al. (2014) demonstrated that Ntf3 acts as a feedback signal between postmitotic neurons and progenitors in the developing mouse neocortex. Our results suggest an additional role for intercellular communication through secreted vesicles in neuronal differentiation and maturation during mouse neocortex development.

### Neuronal EVs promote mESC neural induction
Pluripotent mESCs are guided to alternative specific fates in ectoderm, mesoderm, and endoderm tissues by extrinsic cues

(Young, 2011). EVs may play a regulatory role in stem cell plasticity by supporting cell self-renewal, differentiation, and proliferation (Watt and Huck, 2013). Although there are reports of these effects, little if any molecular mechanistic insight has developed, and in most cases, the effects are observed with cells exposed to crude fractions of sedimentable particles, not isolated membrane vesicles. Here we report that EVs are produced in greater abundance during hormonal and chemically induced neural differentiation of PC12 and N2A cells, and in the differentiation of mouse EBs into neurons, and that these EVs are productively taken up by target cells (Fig. 3, B–G). Buoyant density–purified EVs appear to accelerate aspects of early neural fate conversion of mESCs (Fig. 4, A–E).

In mESCs, cyclin Ds are expressed at a low level, which increases in cells transferred to a serum-free medium (White and Dalton, 2005; Liu et al., 2019). In human ESCs, overexpression of cyclin D induces neuroectoderm differentiation (Pauklin and Vallier, 2013). Here, we found that EVs from neuronal differentiated cells, but not those secreted from undifferentiated cells, stimulated the expression of genes characteristic of mESC neural induction (Figs. 4 C and S4 A). Correspondingly, we found that cyclin D1 was selectively sorted into EVs secreted by differentiating neuronal EVs but not in EVs from undifferentiated PC12 cells, N2A cells, and mESCs, either undifferentiated or in conversion to neural progenitor (ES-D8 EBs; Fig. 5, B–D; and Fig. S5 A). Furthermore, overexpression of EV–cyclin D1 increased the expression of neural markers in mESCs (Fig. 7, J and K). Conversely, EVs from cells depleted of cyclin D1 showed a reduced effect on mESCs (Fig. 7, L and M).

To begin to explore the means by which internalized EVs may influence neuronal gene expression in recipient mESC cells, we applied a proximity labeling approach to detect possible intracellular targets of exogenous cyclin D1. EVs were isolated from donor differentiating N2A cells stably transfected with a cyclin D1–APEX gene fusion (Lobingier et al., 2017). Biotinylated target proteins included two nuclear proteins, Lin28 and nucleolin, which may be involved in the neural promoting effect of EV–cyclin D1. Lin28, for example, is a stem cell maintenance factor (Shyh-Chang and Daley, 2013). EV–cyclin D1 may contribute to the regulation Lin28 and other targets, driving mESC neural conversion.

Our results also raise the question of how proteins contained within EVs may become exposed to the cytoplasm/nucleus of target cells. Although many diverse effects have been attributed to the internalization of proteins and nucleic acids contained within EVs, the existence of a membrane fusion process or membrane channels that would allow such a topological transfer of macromolecules has not yet been demonstrated.

## Materials and methods
### Cell culture, differentiation, and treatment
Rat PC12 cells were maintained in DMEM medium supplemented with 10% horse serum, 5% FBS (GE Healthcare), and 0.1% (vol/vol) penicillin–streptomycin solution. PC12 cells were seeded onto collagen-coated plastic dishes. Differentiated PC12 cells were cultured at a density of $5.5–6.5 \times 10^4/cm^2$ in DMEM medium supplemented with NGF (50 ng/ml; Alomone Labs) and 1% FBS. Mouse N2A cells were cultured in DMEM medium with 10% FBS and 0.1% (vol/vol) penicillin–streptomycin solution. Differentiated N2A cells were cultured at a density of $4–5 \times 10^4/cm^2$ in DMEM medium with RA (10 µM; Sigma-Aldrich, R2625) and 1% FBS. For EV collection, cells were cultured in EV-depleted FBS (System Biosciences, Palo Alto, CA). 14 plates of 150-mm dishes generated 420 ml of conditioned medium harvested from PC12 or N2A cells for each experiment. For the nondifferentiated PC12 or N2A cells, medium was collected after 3 d of culture, when cell confluency reached ~70–80%. For the neuronal differentiation, medium was harvested and replaced every 3 d.

mESCs (R1, gift from Robert Tjian laboratory, University of California, Berkeley, Berkeley, CA) were maintained on N2B27 medium plus 2i (3 µM CHIR99021 and 1 µM PD0325901; Selleckchem) and leukemia inhibitory factor (Millipore Sigma). Dishes were precoated with 0.1% gelatin for 3 h. ESC serum-free monolayer neural progenitor differentiation was performed in N2B27 medium after leukemia inhibitory factor and 2i withdrawal, and ESCs were cultured at a density of $4–5 \times 10^4/cm^2$. For ESC neuronal differentiation, cells were first cultured in suspension in 5% Knockout Serum Replacement medium (KSR; 10828010; Thermo Fisher Scientific) for 8 d to form EBs to achieve neural progenitor status. Then, trypsinized EBs were cultured in poly-D-lysine precoated dishes for another 4 d in N2 medium (Bibel et al., 2007; Zhu et al., 2014). During ES 12-d neuronal differentiation, the medium was changed every 2 d.

For PC12/N2A EV collection, medium was harvested from 3-d-cultured PC12 or N2A cells, and medium from differentiated PC12 or N2A cells was collected every 3 d. For ES EV collection, medium was harvested at 2-d intervals, including 2-d-cultured pluripotent ESCs (ES D0-EV), differentiated ESC-derived EBs cultured for 6–8 d (ES D8-EV), and EB trypsinized neurons for 10–12 d (ES D12-EV).

For EV functional studies, purified EVs from nondifferentiated and differentiated PC12 or N2A cells were added to ESCs in N2B27 medium for a monolayer neural induction process. The medium together with purified EVs were changed every day. EVs were used at ~2–3 $\times 10^9$ particles/ml of N2B27 medium. Particle number was quantified by NanoSight NS300. To measure the efficiency of interaction between EVs and recipient cells, the N2B27 monolayer, but not the KSR EB culture system, was used for recipient mESC differentiation. 1 liter N2B27 medium: 487 ml DMEM/F12, 487 ml neurobasal medium, 10 ml B27 (17504044), 5 ml N2 (17502048), 10 ml L-glutamine (200 mM), 10 ml nonessential amino acids (100×; all from Thermo Fisher Scientific), and 1 ml of 0.1 M β-mercaptoethanol (M3148; Sigma-Aldrich). KSR medium: Glasgow's MEM supplemented with 5% KSR, 2 mM glutamine, 1 mM pyruvate, 0.1 mM nonessential amino acids, and 0.1 mM 2-mercaptoethanol (all from Thermo Fisher Scientific). N2 medium: DMEM/F12 supplemented with N2, 2 mM glutamine, 1 mM pyruvate, 0.1 mM nonessential amino acids, and 0.1 mM 2-mercaptoethanol.

## Differential centrifugation and EV purification

For the two-step EV fractionation (differential velocity centrifugation and linear iodixanol gradient flotation), 420 ml of conditioned medium was harvested from PC12 cells. All subsequent manipulations were performed at 4°C. Cells and large debris were removed by centrifugation in a Sorvall R6+ centrifuge (Thermo Fisher Scientific) at 1,000 $g$ for 15 min followed by 10,000 $g$ for 15 min in 500-ml vessels using a fixed-angle FI-BERlite F14-6×500 y rotor (Thermo Fisher Scientific). The supernatant was centrifuged at 100,000 $g$ (28,000 rpm) for 1.5 h in a SW-28 rotor (Beckman Coulter) with a sucrose cushion. The cushion consisted of 2 ml of 60% sucrose in buffer C (20 mM Tris-HCl, pH 7.4, and 137 mM NaCl). The supernatant was removed carefully, without reaching the cushion, and new conditioned medium was added carefully on top without disturbing the cushion and centrifuged again at 100,000 $g$ (28,000 rpm) for 1.5 h in a SW-28 rotor. The interface between the cushion and the conditioned medium was collected (~3 ml per tube). Approximately 9 ml interface that came from 3xSW28 tubes (25 × 89 mm; Beckman Coulter) was loaded in a SW41 tube (14 × 89 mm; Beckman Coulter) with 0.75 ml of a 60% cushion on the bottom, then centrifuged for 15 h in a SW-41 rotor at 160,000 $g$ (36,000 rpm). The combined interface from the first SW28 sucrose cushion should not exceed a sucrose concentration of 21%, as measured by refractometry, for the second centrifugation in the SW41 to be successful. A clear white band that corresponded to the EV fraction (~1 ml in each SW41 tube) was collected after centrifugation. For purification of EV subpopulations based on their distinct buoyant density, the cushion-sedimented vesicles were collected and mixed with 60% sucrose until to a 4-ml solution of 40% sucrose (in buffer C). The 4-ml solution was then loaded at the bottom of a SW41 tube, and equal amounts (1.5 ml each) of solutions of 25, 20, 15, 10, and 5% iodixanol (Optiprep; diluted in buffer C) were layered on top and centrifuged at 160,000 $g$ (36,000 rpm) for 15 h. Fractions of 400 µl each from top to bottom were taken for evaluation. For immunoblot analysis, the floated fraction samples were mixed with 2× SDS loading buffer (0.125 M Tris-HCl, pH 6.8, 4% SDS, 20% glycerol, 10%-mercaptoethanol, and 0.2% bromophenol blue) and heated at 95°C for 10 min.

For EV purification without resolution of EV subpopulations, 420 ml of conditioned medium was harvested from PC12 or N2A cells. Cells and large debris were removed by stepwise centrifugation at 1,000 $g$ for 15 min, and then 10,000 $g$ for 15 min at 4°C. The supernatant was centrifuged at ~100,000 $g$ (28,000 rpm) for 1.5 h using two SW-28 rotors (Beckman Coulter). The pellet was resuspended with 200 µl PBS, pH 7.4, and diluted in up to ~5 ml of PBS, followed by centrifugation at ~150,000 $g$ (38,500 rpm) in an SW-55 rotor (Beckman Coulter). Washed pellet material was then resuspended in 100 µl PBS as in the first centrifugation step, and 900 µl of 60% sucrose (in buffer C) was added and mixed, and 2 ml of 40% sucrose (in buffer C) and 1 ml of 20% sucrose (in buffer C) were sequentially overlaid. The SW55 tubes (13 × 51 mm; Beckman Coulter) were centrifuged at ~150,000 $g$ (38,500 rpm) for 16 h in an SW-55 rotor. The 20/40% interface was harvested and washed once with PBS in an SW-55 rotor. The sedimented EV fraction was resuspended in 100 µl PBS for further analysis.

## Nanoparticle tracking analysis

EV sizes and quantities were estimated using the NanoSight NS300 instrument equipped with a 405-nm laser (Malvern Instruments), analyzed in the scatter mode. Silica 100-nm microspheres (Polysciences) served as a control to check instrument performance. Vesicles collected as described above were diluted 1,000× with filtered PBS (0.02 µm; Whatman). The samples were introduced into the chamber automatically, at a constant flow rate of 50 during five repeats of 60-s captures at camera level 13 in scatter mode with Nanosight NTA 3.1 software (Malvern Instruments). The size was estimated at detection threshold 5 using Nanosight NTA 3.1, after which "experiment summary" and "particle data" were exported. Particle numbers in each size category were calculated from the particle data, in which "true" particles with track length >3 were pooled, binned, and counted with Excel (Microsoft).

## PKH67 labeling

EVs were labeled with fluorescent dye PKH-67 using the PKH-67 labeling kit (Sigma-Aldrich). Briefly, 5 × 10$^{10}$ EVs were resuspended in 100 µl PBS and mixed with 100 µl of PHK67 dye diluted in diluent C (4 µl of the PKH67 dye solution to 1 ml of diluent C) for 2 min, followed by continuous mixing for 30 s by gentle pipetting. Excess dye was quenched by adding 100 µl of 10% BSA in PBS. This mixture was diluted with 4.5 ml of PBS and centrifuged at 150,000 $g$ (38,500 rpm) for 1 h to sediment the PKH-67–labeled EVs. The EV pellet was further washed twice with PBS by centrifugation at 150,000 $g$ for 60 min to remove any free dye, and the final EV pellet was resuspended in 100 µl PBS and used for uptake studies.

## Proteinase K protection assay

The EVs fractionated by differential centrifugation and sucrose flotation were aliquoted into 20 µl of PBS or PBS containing indicated concentrations of proteinase K (proteinase K was dissolved in TBS, pH 7.4, 5 mM CaCl$_2$, and 50% glycerol) on ice for 20 min, and then treated with or without 1% Triton X-100 on ice for 10 min. The reactions were stopped by sequentially adding PMSF to final concentration of 5 mM, and aliquots were mixed with 2× SDS loading buffer followed by at 95°C for 10 min. Samples were processed for SDS-PAGE and evaluated by immunoblot.

## Immunoblotting

Standard immunoblotting procedures were followed. In brief, samples were heated at 95°C for 10 min, resolved on 4–20% polyacrylamide gels (15-well, Invitrogen; 26-well, Bio-Rad), and transferred to polyvinylidene fluoride (EMD Millipore). The polyvinylidene fluoride membrane was incubated with antibodies (primary for 4°C overnight and secondary for 1 h at RT), and bound antibodies were visualized by the enhanced chemiluminescence method (Thermo Fisher Scientific) on a ChemiDoc Imaging System (Bio-Rad) with ImageLab software v4.0 (Bio-Rad). The following antibodies were used: rabbit anti-cyclin D1, anti-cyclin A2, anti-CDK4, anti-CD9, anti-Flotillin 2, anti-p21, anti-p27, anti-p57, anti-nucleolin, anti-Ngn2, anti-integrin (ab134175, ab181591, ab199728, ab92726, ab96507, ab109199, ab32034,

ab75974, ab129200, ab109236, and ab131055; Abcam), rabbit anti-CDK6 (GTX103992; GeneTex), rabbit anti–cyclin B1, anti–cyclin E1, anti–phospho-Rb (4138T, 20808S, and 9307; Cell Signaling Technology), rabbit anti-GFP (NC9589665; Fisher Scientific), rabbit anti-Lin28 (11724-1-AP; Proteintech); mouse anti-actin (ab8224; Abcam), mouse anti–cyclin D2, anti–cyclin D3, anti-Alix, anti-CD81 (sc-376676, sc-6283, sc-53540, and sc-166029; Santa Cruz Biotechnology), mouse anti-GM130 (610823; BD Biosciences), mouse anti-Tsg101 (GTX70255; GeneTex); rat anti-CD63 (clone R5G2, LS-C179520; LSBIO), rat anti-Hsc70 (ab19136; Abcam), HRP-conjugated streptavidin (N100; Thermo Fisher Scientific), GFP protein recombinant (PRO-687; ProSpec), NGF neutralizing antibody (MAB256-100; R&D Systems), HRP-linked IgG from mouse and rabbit (NXA931,45000682; Fisher Scientific), and HRP-linked IgG from rat (A5795; Sigma-Aldrich).

## RNA preparation and qPCR analysis
Total RNA was extracted from cells using TRIzol reagent (Invitrogen). 2.5 μg RNA was reverse transcribed by superScript III reverse transcription (Thermo Fisher Scientific) according to the manufacturer's instructions. qPCR was performed using TaqMan Universal PCR Master Mix on an ABI-7900 real-time PCR system (Thermo Fisher Scientific). Primers for qPCR analysis are listed in Data S1.

## Negative staining and visualization of EVs by EM
The EVs collected as described were resuspended in 1% glutaraldehyde, spread onto glow-discharged Formvar-coated copper mesh grids (Electron Microscopy Sciences), and stained with 2% uranyl acetate for 2 min. Excess staining solution was removed by blotting with filter paper. After drying, grids were imaged at 120 kV using a Tecnai 12 Transmission Electron Microscope (FEI) housed in the Electron Microscopy Laboratory at the University of California, Berkeley.

## Immunofluorescence
Cells growing on Falcon 4-well Culture Slides (Corning) were fixed in 4% PFA for 30 min at RT, washed five times with PBS, and incubated with blocking buffer (PBS containing 0.1% Triton X-100 and 0.5% BSA) at RT for 1 h. Cells were incubated with primary antibody at 4°C overnight, washed five times with PBS, and incubated with secondary antibody for 1 h at RT. The following antibodies were used: mouse anti-nestin (ab6142; Abcam); rabbit anti-Pax6 (ab195045; Abcam); rabbit monoclonal anti-CD9 (ab92726; Abcam); rabbit anti-GFP (NC9589665; Fisher Scientific); Alexa Fluor 488 donkey anti-mouse IgG (A-11001; Invitrogen); and Alexa Fluor 568 goat anti-rabbit IgG (A-10042; Invitrogen). Antibody incubations were followed by five washes with PBS. Coverslips were mounted in ProLong-Gold antifade mountant with DAPI (Thermo Fisher Scientific) overnight before imaging. Images were acquired using Zen 2010 software on an LSM 710 confocal microscope system (Zeiss) and Plan-Apochromat 100×, 1.4-NA objectives.

## Coimmunoprecipitation (coIP) assay
For EV CD9 immunoprecipitation, ~5 × 10^10 EVs collected from the two steps of purification were diluted into 500 μl PBS and 2 μg rabbit monoclonal anti-CD9 (ab92726; Abcam), or rabbit IgG (Fisher Scientific) was added and mixed by rotation overnight at 4°C. Magvigen protein-A/G–conjugated magnetic beads (30 μl; Nvigen) were then added to the EV/antibody mixture and mixed by rotation for 2 h at 4°C. Beads with bound EVs were washed three times in 1 ml PBS, and protein was extracted using with 2× SDS loading buffer. The samples were heated at 95°C for 10 min and analyzed by SDS-PAGE and immunoblot. Immunoprecipitation of exosomal cyclin D1 was performed as with the immunoprecipitation of CD9, except that the purified EVs were diluted and washed in coIP buffer, containing 50 mM Tris HCl, pH 7.4, 150 mM NaCl, 1 mM EDTA, 1% Nonidet P-40, 10% glycerol, 1 mM PMSF, 1 mM DTT, and 1× proteinase inhibitors [Roche]). Anti–cyclin D1 antibody (2 μg; ab134175; Abcam) or rabbit IgG (Fisher Scientific) was added.

For coIP of proteins in a cell lysate, the suspended cells were lysed at 4°C with coIP buffer. Lysates were incubated with required antibodies at 4°C overnight and then with 30 μl Dynabeads Protein G (Thermo Fisher Scientific) for another 2 h. The immunocomplexes were centrifuged and washed three times with cold coIP buffer and once with 50 mM Tris HCl, pH 7.4, in the absence of proteinase inhibitors. The proteins were released from beads by heating to 95°C in SDS sample buffer, and the samples were analyzed by immunoblot. The following antibodies were used for coIP: anti-Flag (1:1,000; Sigma-Aldrich) and anti-HA (1:1,000; Cell Signaling Technology).

## BrdU assay
A BrdU cell proliferation assay was conducted according to the supplier's instructions (Cell Signaling Technology). Briefly, the purified EV fraction was added or not to mESC cultures at the onset of the neural induction process, in N2B27 medium for 4 d. After 3 d of mESC differentiation, cells were incubated for a further 24 h with BrdU. Cells were then fixed and incubated with 1× detection antibody, washed three times with wash buffer, and incubated with HRP-conjugated antibody for 0.5 h at RT. Fixed and labeled cells were then incubated with the 3,3', 5,5"-tetramethylbenzidine substrate, and reactions were terminated with the stop solution. Outcomes were recorded by absorbance at 450 nm.

## Plasmid construction
The plasmids encoding CDK4-Flag, Hsc70-HA, and Hsp90-HA were purchased from Sino Biology. The cyclin D1–Flag plasmid was generated by PCR insertion of cyclin D1 into the p3XFLAG-CMV-14 expression vector (Sigma-Aldrich). The XPack-GFP plasmid was generated by inserting GFP from pEGFP-N1 (BD Biosciences Clontech) into the XPack CMV-XP-MCS-EF1α-Puro Cloning Lentivector (System Biosciences). The cyclin D1–APEX plasmids were constructed by combining the PCR fragment of cyclin D1 from the cyclin D1–Flag plasmid and APEX from pcDNA3 APEX-nes (49386; Addgene) into XPack CMV constructs (System Biosciences). The cyclin D1–GFP plasmids were constructed by combining the PCR fragment of cyclin D1 from the cyclin D1–Flag plasmid and GFP from pEGFP-N1 (BD Biosciences Clontech) into XPack CMV constructs (System Biosciences). For cyclin D1 overexpression, the cyclin D1 was generated by PCR

insertion into the XPack CMV constructs (System Biosciences). Hsc70D10N was generated by PCR-based site-directed mutagenesis using the QuikChange II XL Site-Directed Mutagenesis Kit (Agilent Technologies). Cyclin D1 KO was conducted by CRISPR/Cas9 genome editing (Xie et al., 2016). A pX330-based plasmid expressing GFP was used to clone the gRNAs targeting cyclin D1. Three CRISPR gRNAs targeting the gene were selected using the CRISPR design tool; gRNAs targeting exon 1 were selected. Primers for gRNAs are listed in Data S1.

### CRISPR interference

N2A cells expressing UCOE-EF1α-dCas9-BFP-KRAB were obtained by lentivirus transduction (Gilbert et al., 2014; 102244; Addgene). Cells were sorted for the BFP signal 3 d after transduction, and selected cells were expanded by growth for a few generations and then frozen and stored as parental cells (referred to as dCas9). Sequences for gRNAs targeting the promoter of the genes of interest were selected based on data by Horlbeck et al. (2016). gRNAs were cloned in plasmid pU6-sgRNA EF1Alpha-puro-T2A-BFP (Gilbert et al., 2014; 60955; Addgene). The top two gRNAs from the V.2 library (Horlbeck et al., 2016) were chosen to transduce the parental dCas9 cells. Posttransduction cells were selected with 2 μg/ml puromycin for 3 d.

### APEX reaction and biotinylated protein capture
#### APEX reaction in cells

Cylin D1–APEX N2A cells were used to capture the proximity labeling reaction or collect purified cylin D1–APEX EVs. For the APEX reaction in receipt mESCs, cylin D1–APEX EVs were incubated with mESCs in N2B27 medium for 2 d at a concentration of ~5 × 10⁹ EVs/ml medium. APEX proximity labeling was conducted as described previously (Hung et al., 2016). Biotin-phenol (500 μM) was preincubated with cells for 30 min at 37°C. Immediately before use, 1 mM (0.003%) $H_2O_2$ (Thermo Fisher Scientific) was spiked into the medium for the 1-min labeling reaction at RT. The reaction was then quenched immediately by three thorough washes with RT quencher solution, containing 10 mM sodium ascorbate (Sigma-Aldrich), 5 mM Trolox (Sigma-Aldrich), and 10 mM sodium azide (Sigma-Aldrich) in Dulbecco's PBS (Thermo Fisher Scientific). Cells were lysed in radioimmunoprecipitation assay (RIPA) medium (50 mM Tris, 150 mM NaCl, 1% Triton X-100, 0.5% deoxycholate, and 0.1% SDS, pH 7.4) supplemented with 10 mM sodium ascorbate, 1 mM sodium azide, 1 mM Trolox, 1 mM DTT, and protease inhibitors (Roche). The whole-cell lysate was combined with loading buffer, heated at 95°C for 10 min, and resolved by SDS-PAGE. Biotinylated proteins were evaluated by blotting with streptavidin-HRP (21130; Thermo Fisher Scientific).

#### APEX reaction in EVs

500 μM biotin-phenol was incubated with purified EVs for 30 min at 37°C in a total mixture volume of <50 μl. The mixture was removed to an SW 41 ultracentrifuge tube, and APEX labeling was initiated by addition of 1 mM $H_2O_2$. After 1 min, 12 ml quencher solution was added, and EVs were sedimented and washed with the quencher solution twice by centrifugation at 110,000 g (31,500 rpm) for 1 h. The pellet fraction was suspended

in 40 μl PBS and mixed with SDS-loading buffer in preparation for SDS-PAGE and blotting.

### Preparation of APEX-labeled proteins for blotting or MS

Samples in RIPA were briefly sonicated in a bath sonicator (S220; Covaris) and centrifuged at 10,000 g for 10 min. The supernatant fraction (800 μl) was applied to 40 μl streptavidin-agarose beads (Sigma-Aldrich) followed by rotation overnight at 4°C. Streptavidin-agarose beads were washed two times in RIPA lysis buffer, once with 1 M KCl, once with 0.1 M $Na_2CO_3$, and once with 2 M urea in 10 mM Tris-HCl, pH 7.4. Biotinylated proteins were eluted from the beads by heating the sample in 4× SDS loading buffer supplemented with 2 mM biotin and 20 mM DTT for 10 min at 95°C. Streptavidin-HRP blotting or MS was used to identify the biotinylated proteins. For MS, heated samples were electrophoresed in a 4–20% acrylamide Tris-glycine gradient gel (Life Technologies) for ~3 min. The proteins were stained with Coomassie, and stained bands were excised from the gel with a fresh razor blade. Samples were submitted to the Taplin Mass Spectrometry Facility at Harvard Medical School (Cambridge, MA) for in-gel tryptic digestion of proteins followed by liquid chromatography and MS analysis according to their standards.

### Statistical analysis

Statistical analysis was performed using Prism (GraphPad Software). Groups were compared using Student's $t$ test. The values represent the mean ± SD from two or three independent experiments. NS, $P > 0.05$; *, $P < 0.05$; and **, $P < 0.01$).

### Online supplemental material

Fig. S1 shows vesicle size and protein concentration quantification of different subpopulation of the EVs. Fig. S2 shows neuronal differentiation of N2A cells and mESCs. Fig. S3 shows that RA-induced EVs promote mESC neural fate commitment. Fig. S4 shows that cyclin D1 is enriched in EVs during N2A neurogenesis. Fig. S5 shows MS analysis of RA-EV and cyclin D1–KO EV. Data S1 shows raw data of the MS analysis and all primer lists.

## Acknowledgments

We thank Amita Gorur for help and advice on EM. We thank the staff at University of California, Berkeley shared facilities, including Bob Lesch and Alison Killilea. We thank Criss Hartzell, David Melville, Arup Indra, and Xiaoman Liu for reading the manuscript.

L. Song was supported in part by the Tang Family Fellowship and as a Howard Hughes Medical Institute Associate. R. Schekman is an Investigator of the Howard Hughes Medical Institute and a Senior Fellow of the University of California, Berkeley Miller Institute for Basic Research in Science.

The authors declare no competing financial interests.

Author contributions: L. Song and R. Schekman conceptualized the experiments and study. L. Song and X. Tian carried out research experiments. L. Song and R. Schekman analyzed data. L. Song and R. Schekman prepared the original draft and revised the manuscript.

Submitted: 14 January 2021

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

# Supplemental material

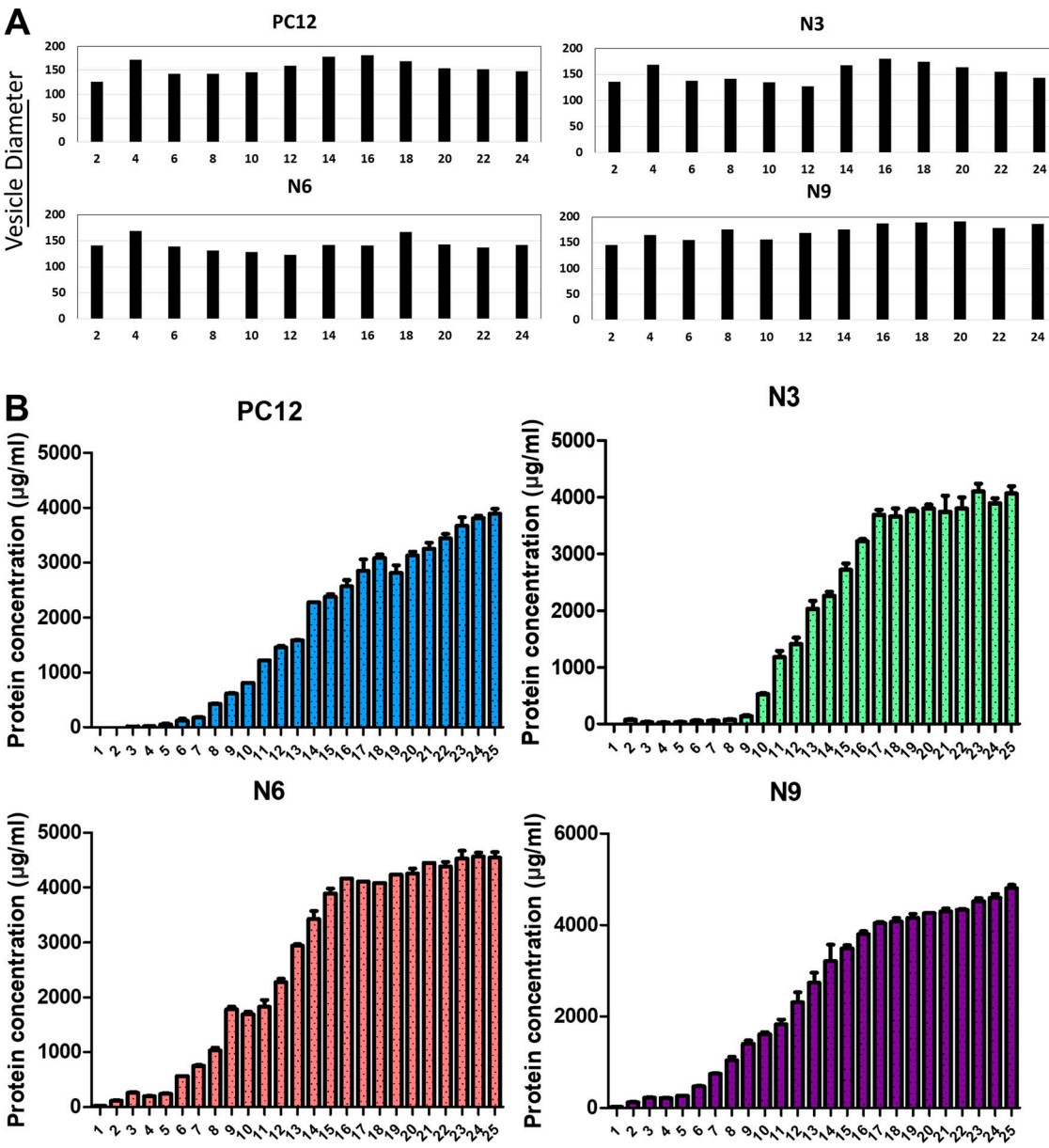

Figure S1. **Vesicle size and protein concentration quantification of different subpopulations of EVs. (A)** The size of EVs in different fractions and at different differentiation time points was measured with a NanoSight particle tracking device. **(B)** The protein concentration of EVs in different fractions and at different differentiation time points was measured using a microBCA kit. Data plotted are from two independent experiments, each with triplicate qPCR reactions; error bars represent SD from independent samples.

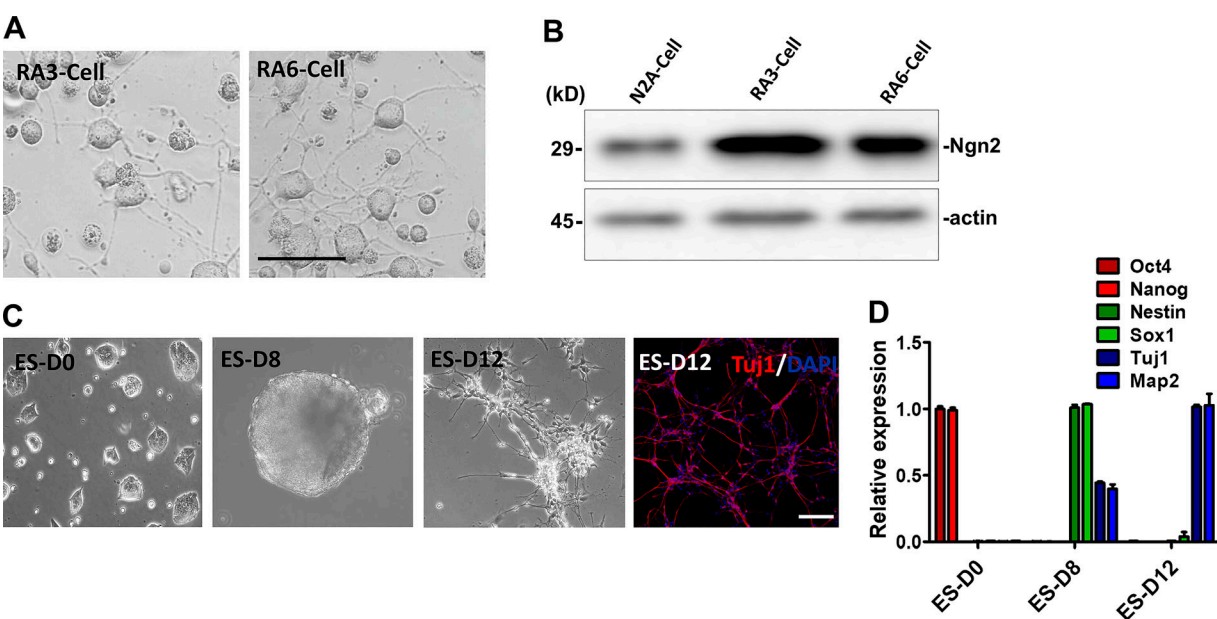

Figure S2. **Neuronal differentiation of N2A cells and mESCs. (A)** Cellular morphology of N2A cells cultured in low-serum medium with RA (10 μM) for 3 and 6 d. Scale bars, 50 μm. **(B)** Immunoblots of Ngn2 and actin in N2A cells untreated and cultured in RA (10 μM) for 3 and 6 d. **(C)** Cellular morphology of undifferentiated pluripotent ESCs (ES-D0), EBs of ES differentiated 8 d in KSR medium (ES-D8), and EBs trypsinized in N2 medium for an additional 4 d (ES-D12). Immunostaining of ES-D12 neuronal cells with Tuj1 (red, Alexa Fluor 568) and DAPI (blue). Scale bars, 200 μm. **(D)** The expression of pluripotent markers Oct4 and Nanog, neural progenitor markers Nestin and Sox1, and neuronal markers Tuj1 and Map2 was determined by RT-PCR. Samples were collected as described in C. Data plotted are from two independent experiments, each with triplicate qPCR reactions; error bars represent SD from independent samples.

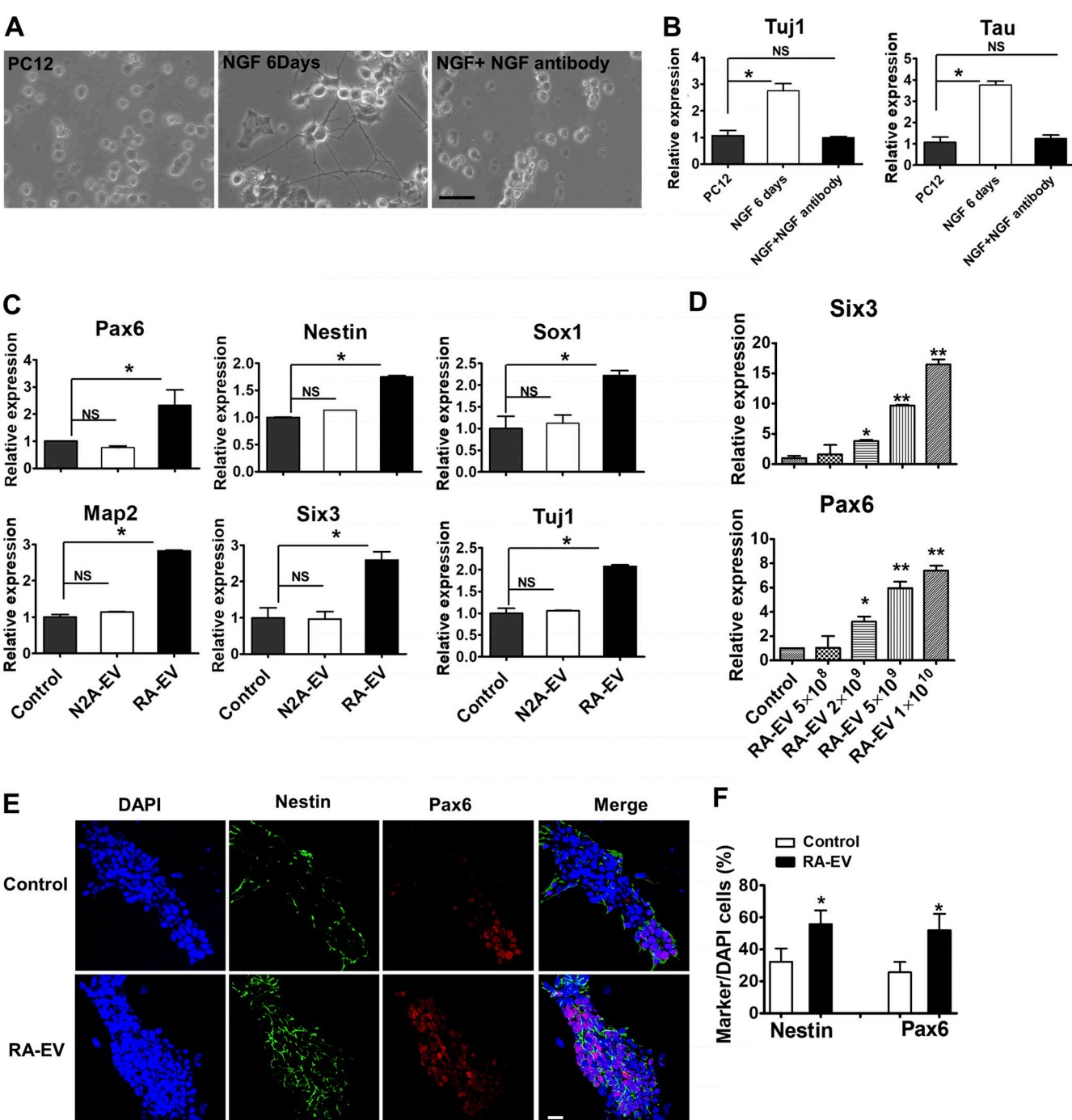

Figure S3. **RA-induced EVs promote mESC neural fate commitment. (A)** The cellular morphology of PC12 cells untreated or treated with NGF (50 ng/ml) without or with NGF neutralizing antibody (500 ng/ml) for 6 d. Scale bars, 50 μm. **(B)** Gene expression analysis of *Tuj1* and *Tau* in PC12 cells untreated and treated with NGF (50 ng/ml) without or with NGF neutralizing antibody (500 ng/ml) for 6 d. The values represent the mean ± SD, from two independent experiments (*, P < 0.05; NS, P > 0.05). Error bars represent SD from independent samples. **(C)** Gene expression analysis of mESCs treated with EVs purified from N2A cells (N2A-EV) or EVs purified from 6-d RA-treated cells (RA-EV). The values represent the mean ± SD, from two independent experiments (*, P < 0.05; NS, P > 0.05). Error bars represent SD from independent samples. **(D)** Gene expression of *Six3* and *Pax6* in mESCs treated with different doses of RA-EV. The values represent the mean ± SD, from three independent experiments (*, P < 0.05; **, P < 0.01). Error bars represent SD from independent samples. **(E)** Immunostaining of mESCs described in A. Cells were stained with Nestin (green, Alexa Fluor 488) and Pax6 antibodies (red, Alexa Fluor 568). Scale bars, 25 μm. **(F)** Quantitative analysis of the staining in C. The values represent the mean ± SD, from three independent experiments (*, P < 0.05). Error bars represent SD from independent samples.

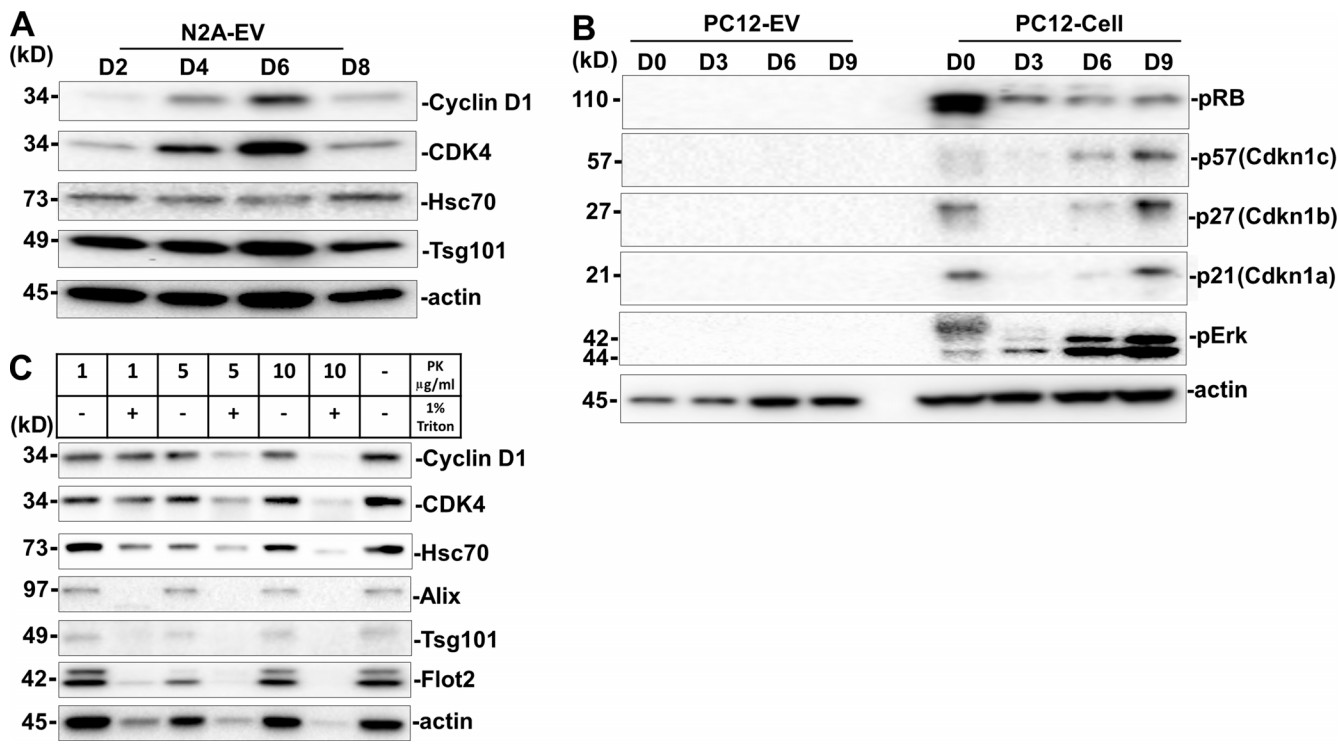

Figure S4. **Cyclin D1 is enriched in EVs during N2A neurogenesis. (A)** Immunoblots of cyclin D, CDK4, Hsc70, Tsg101, and actin of EVs from RA-induced N2A cells for 2, 4, 6, and 8 d (D2, D4, D6, and D8). **(B)** Immunoblots of pRB, p57, p27, p21, pErk, and actin in differentiated PC12 cells and EVs. **(C)** Immunoblots of cyclin D1, CDK4, and multiple EV markers from the N6-EVs treated with different concentrations of proteinase K (PK), with or without 1% Triton X-100.

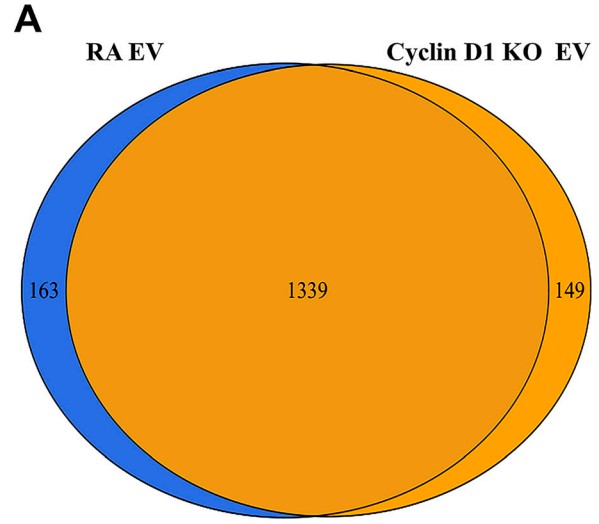

Figure S5. **MS analysis of RA-EV and cyclin D1–KO EV. (A)** Whole-protein profile of RA-EV and cyclin D1–KO EV was analyzed by MS. The proteome (1,339 proteins) overlapped extensively in these two preparations. The list of the EV proteins shown in Data S1.

**Provided online is one dataset. Data S1 shows raw data of the MS analysis and all primer lists.**

