## [Peer Review File · The Journal of Cell Biology]

Extracellular vesicles from neurons promote neural induction of stem cells through cyclinD1

Lu Song, Xinran Tian, and Randy Schekman

Corresponding Author(s): Randy Schekman, University of California, Berkeley

Review Timeline:

Submission Date:	2021-01-14
Editorial Decision:	2021-02-24
Revision Received:	2021-05-08
Editorial Decision:	2021-05-14
Revision Received:	2021-05-25

Monitoring Editor: Louis Reichardt

Scientific Editor: Melina Casadio

Transaction Report:

DOI: <https://doi.org/10.1083/jcb.202101075>

February 24, 2021

Re: JCB manuscript #202101075

Dr. Randy Schekman
University of California, Berkeley
Department of Molecular and Cell Biology University of California at Berkeley 482 Li Ka Shing Center
#3370
Berkeley, CA 94720-3202

Dear Randy,

Thank you for submitting your manuscript entitled "Extracellular vesicles from neuronal cells promote neural induction of mESCs through cyclinD1" to Journal of Cell Biology. I am attaching to this letter the evaluations by three reviewers of the manuscript. The reviewers have several concerns: first that the argument that the vesicles that you are characterizing are exosomes and not vesicles derived from the plasmalemma (e.g., Kowal et al., 2016). Secondly, a limited assessment of novelty because of prior work (Sharma et al., 2019; Pauklin et al., 2016). Third, some reservations about relevance in view of the absence of any demonstration of importance in organoids or the intact developing brain, especially relevant in view of the first reviewer's comments regarding levels of cyclin D1 expression in vivo in neurons as well as this reviewer's further comments about more robust expression of cyclin D1 in other brain cell types as well as some concerns about the limited association of differentiating neurons with the apical and intermediate progenitors in the murine brain (I am not overlooking the more positive assessment of this reviewer compared to those of the others in highlighting these comments). Overall, this left me additionally with some concern about the wisdom of using the PC12 and N2A lines vs. primary neurons in your experiments.

After reviewing your manuscript and the assessments of each reviewer, I do not think this manuscript can be accepted by this journal in its current form and that the very extensive numbers of additional experiments that would be needed to raise its significance and novelty to this journal's expectations are too high to encourage revision. Consequently, I must reject the manuscript with a recommendation that you submit it elsewhere.

I am a bit chagrined to have to send someone with your stature a decision letter of this nature. I also hope that the decision does not discourage your laboratory members who have invested so much effort in this study.

Although we regret that we are not able to consider your manuscript further, we have discussed your manuscript with the editors of Life Science Alliance (<http://www.life-science-alliance.org/>) and they would like to offer publication pending text edits to respond to the reviewers' comments and in particular, edits to address the 'exosome' nomenclature point (using 'EV' instead) as pointed out by Reviewer #2 and the other minor points raised by all reviewers. LSA is our academic editor-led, open-access journal launched as a collaboration between RUP, EMBO Press, and Cold Spring Harbor Lab Press. You can use the link below to initiate an immediate transfer of your manuscript files and reviewer comments to LSA.

Link Not Available

Thank you for your interest in the Journal of Cell Biology.

With best personal regards,

Louis Reichardt, PhD
Editor, Journal of Cell Biology

Melina Casadio, PhD
Senior Scientific Editor, Journal of Cell Biology

Reviewer #1 (Comments to the Authors (Required)):

Manuscript by Song et al investigate the role of extracellular vesicles (EVs) in neuronal differentiation. The main thesis behind this work is that cyclinD1 containing EVs are derived from neurons and can promote neuronal induction and differentiation among undifferentiated cells. The study is extremely well executed and very interesting, and opens up a lot of questions about basic cell biology of exosome sorting, neuronal differentiation, and more broadly fate commitment. I think that the manuscript is appropriate for publication after addressing a few minor comments.

In Figure 5A, the authors validate the presence of cyclin D1-3 in exosomes collected from NGF induced differentiation off PC12 cells. The authors say in the paper that they only detect D1 and D2 but not D3, but in their image it appears as though there may be D3 as well. I am confused by the disconnect between the data presented and the conclusion.

"Next, to examine the contribution of exosomal cyclin D1 to mESC neural commitment, we generated cyclin D1-overexpressing N2A cells by us of a lentivirus vector." Should read "by use of a lentivirus".

In the experiment overexpressing cyclin D1, was the control condition devoid of any lentivirus, or was a control virus used. Although I am not aware of any literature showing that virus infection affects the differentiation process, it would be nice if the authors used empty virus as a control.

I enjoyed reading the discussion section of the paper, I think that the authors put forward some very interesting ideas for future exploration. One issue which I am a little concerned about is that in the developing cerebral cortex, neurons are located some distance away from the neural progenitor cells. Whether this kind of mechanism could be sufficiently efficient to be transferred between neurons and radial glia fibers is not known. A related issue is that work from Victor Tarabykin showed another mechanism mediated by Ntf3 also seems to promote neuronal differentiation. I am not sure if such mechanisms could be independent or redundant, but it would be nice to see some discussion.

Finally, I think it is worth pointing out that cyclin D1 is not very robustly transcribed in neurons, and there are other cell types which do express it, such as astrocytes and OPCs. I think it would be valuable if the authors noted this and further elaborated or at least mentioned that although they primarily focused on neuronal lineage, there may be other effects affecting other lineages.

Reviewer #2 (Comments to the Authors (Required)):

This work by Song et al describes the modification of release pattern of small extracellular vesicles (EVs) by neuronal cells (PC12 and N2A) upon differentiation, and an effect of the differentiated cell-derived EVs to induce neural differentiation of murine embryonic stem cells. The authors identify a nuclear protein, cyclin-D1 in the active EVs, and show that it is an important contributor to the mESC induction of differentiation. Through an original approach of APEX-tagging of cyclin-D1, they also identify Hsc70 as a molecule important to target cyclin-D1 to EVs.

The article on a whole is well performed and the conclusions supported by the data. The effect of differentiated neural cell-derived EVs on Embryonic stem cells is probably novel (although I do not know the field of neural development and stem cells enough to be 100% sure), although the relevance to the in vivo situation and physiological consequences are unclear to me. The observation of Cyclin-D1 in EVs and the role of Hsc70 in it are novel and interesting, although the actual cell biology mechanism of transfer of this particular nuclear protein (and not others of the same family) into EVs is not really explained.

A major weakness of the article (but easily solved) is the use of the term "exosomes" throughout, which is not supported by actual demonstration that the EVs analysed (i.e. the EVs that induce mESC differentiation, and the ones that contain cyclin-D1) are specifically formed in MVBs and thus correspond to bona fide exosomes. The authors seem to base this interpretation of the wrong assumption that EVs that float into a iodixanol gradient at the 20-40% interface are exosomes (, rather than PM-derived EVs, and that CD9 is a tetraspanin enriched in MVBs. None are true: other authors have shown that both EVs enriched and not enriched in endosomal components are recovered at such densities, and that enrichment in late endosomal components is not a feature of CD9-bearing EVs (Kowal 2016 # 26858453) . In truth, CD9 is generally expressed at the plasma membrane, and not enriched in MVBs, as opposed to CD63 which is primarily in MVBs. Thus CD9 cannot be considered as an exosome marker. In the same study by Kowal et al, flotillin and Hsc70 were also found in all types of EVs, not specifically in exosomes. The authors must therefore change nomenclature, and use the term EVs throughout. Unless, they want to try to prove that the CyclinD1-containing EVs are exosomes (which would be an interesting message, for a cell biology journal). In that case, they must look for localization of cyclin-D1 (as compared to another Cyclin that is not recovered in EVs) and Hsc70 in the cells: are they colocalized in MVBs rather than in another membrane location? Are the cyclin-D1-EV components, as can be identified by the APEX approach, particularly enriched in late endosomal components? Note that ESCRT proteins are involved in budding both in MVBs and at the plasma membrane (see for instance Hurley 2015 # 26311197), thus their presence in EVs does not prove an exosomal nature.

Apart from this comment, quantification or controls are missing in a few experiments:

In Fig5E showing proteinase K protection assay: a positive control of efficacy of proteinase K (in the absence of triton) must be shown, with digestion of a surface-exposed molecule (for instance CD9, or an integrin).

In Fig5F: the authors perform IP anti CD9, and show the flow through (FL), however, since they recover nothing in FL, even when using a control IgG for IP, where everything should be in Flow-through, these results show that the IP is not efficient.

In Fig4C, they show a good control: anti-NGF antibody that does not block EV-mediated effect. However, a condition with recNGF + anti-NGF should be shown, to show efficiency of the antibody.

In Fig7F, specific APEX-dependent biotinylation of proteins in target cells is not demonstrated, because a control of cells exposed to EVs from regular CyclinD1 (i.e. not-APEX fused)- expressing cells is not performed.

Other comments:

In figure 1D, changes of the density of some EV-associated markers upon neural cell differentiation

is interesting, but their meaning and relevance is difficult to interpret. The authors could at least try to measure the size of the EVs recovered at the different densities and time points, and they should also have determined whether the changed pattern of EV release also induces changes in release of larger or denser EVs recovered in the first steps of centrifugation.

In figures 2G, 2I and 5C: quantifications of signals in WB is relative to what?

Position of molecular weight markers must be indicated in all WB. The CD63 sharp band shown in Figure 2F suggests that the antibody used is not specific: CD63 is a highly glycosylated protein, showing a very smeared pattern in western blots, and is reliably detected in non-reducing conditions only. The Santa Cruz antibody used here is not satisfying. A good rat anti-mouse CD63 is the monoclonal R5G2.

Reviewer #3 (Comments to the Authors (Required)):

Song et al. demonstrate a striking increase in exosome production during neuronal differentiation of PC12 and N2A cell lines. Interestingly, the exosomes isolated from differentiated PC12 and N2A cells, but not those from control (nondifferentiated) cells, promoted expression of neural markers in mouse embryonic stem cells (ESCs) maintained under a differentiation-inducing serum-free condition. The authors then show that cyclin D1 was selectively sorted into the luminal interior of exosomes produced by differentiating PC12 and N2A cells. With the use of APEX proximity labeling and co-immunoprecipitation methods, they also found that Hsc70 associates with cyclin D1 and promotes its sorting into exosomes in differentiated N2A cells. Exosome-derived (APEX-fused) cyclin D1 was found to associate with several proteins including Lin28 and nucleolin after its incorporation into ESCs. Importantly, overexpression or depletion of cyclin D1 in N2A cells promoted and attenuated the effect of N2A cell-derived exosomes on neural differentiation of ESCs, respectively.

Major comments

This is an interesting and solid study describing a new mode of cell-cell communication mediated by exosome-derived cyclin D1 and resulting in induction of neural differentiation. Unfortunately, a previous study (Sharma et al. 2019) has already demonstrated a neural differentiation-inducing function of exosomes obtained from developing neural cultures (human iPSC-derived neural cultures as well as rat primary neural cultures), which diminishes the novelty of the present study—despite the crude nature of the exosome preparations used in the previous report (as pointed out by the present authors). Moreover, the role of cyclin D1 in neuroectoderm induction in ESCs has also previously been described (Pauklin et al. 2016). The conceptual advance made by the present study is thus limited—although this would be mitigated, for example, if the authors can show whether or how the mechanism of neural differentiation in ESCs induced by exosome-derived cyclin D1 differs from that induced by cell-intrinsic cyclin D1. Demonstration of an *in vivo* context in which exosome-derived cyclin D1 actually promotes neural differentiation would also be important.

Specific comments

1. The results shown in Figure 7L and 7M are important in supporting the main conclusion and ruling out a contribution of contaminating (nonexosomal) factors in the prepared exosomal fractions. Other than cyclin D1, is there any difference between the contents of RA Exo and Cyclin D1-KO Exo fractions? Also, counting of marker-positive cells should be performed in Figure 7M in order to assess neural differentiation.

2. Is there any evidence that the treatment with neutralizing antibodies to NGF was sufficient for

inactivation of NGF in Figure 4C?

3. Is Hsc70 in N2A cells necessary for neural differentiation-inducing activity of RA Exo, given that it mediates cyclin D1 packaging into exosomes?

Rockefeller University Press
950 Third Ave., 2nd Floor
New York, NY 10022
jcellbiol@rockefeller.edu

May 8, 2021

Re: MS 202101075

Dear editor,

Based on the editor and reviewers' comments and suggestions, we have repeated several of our experiments using mESCs differentiated into embryoid bodies and then into Tuj1+ neurons (new data of Fig. 2F; Fig. S2 C, D; Fig. 4F). We performed more experiments, added main data and supplementary data for Figure 1 (Fig. S1 A, B), Figure 4 (Fig. S4 A, B), Figure 6 (Fig. 6 I, J) Figure 7 (Fig. S7 A), and revised the manuscript accordingly. Please find the point-by-point response to the concerns and comments of the reviewers below. We believe that all the questions raised by the editor and reviewers have been clearly addressed. We hope that our manuscript now is suitable to be accepted in *JCB*.

I greatly appreciate you and the reviewers for your time and efforts to make our manuscript better.

Sincerely yours,

Randy Schekman,
University Professor
Department of Molecular and Cell Biology,
and Investigator, Howard Hughes Medical Institute
University of California, Berkeley

Reviewer #1 (Comments to the Authors (Required)):

Manuscript by Song et al investigate the role of extracellular vesicles (EVs) in neuronal differentiation. The main thesis behind this work is that cyclinD1 containing EVs are derived from neurons and can promote neuronal induction and differentiation among undifferentiated cells. The study is extremely well executed and very interesting, and opens up a lot of questions about basic cell biology of exosome sorting, neuronal differentiation, and more broadly fate commitment. I think that the manuscript is appropriate for publication after addressing a few minor comments.

In Figure 5A, the authors validate the presence of cyclin D1-3 in exosomes collected from NGF induced differentiation off PC12 cells. The authors say in the paper that they only detect D1 and D2 but not D3, but In their image it appears as though there may be D3 as well. I am confused by the disconnect between the data presented and the conclusion.

Thanks for the questions. We found cyclin D1-3 were gradually up-regulated in NGF -induced PC12 cells (Fig 5A). In isolated EVs, we found cyclin D1 and D2 but not cyclinD3 (not shown) (Fig 5B).

"Next, to examine the contribution of exosomal cyclin D1 to mESC neural commitment, we generated cyclin D1-overexpressing N2A cells by us of a lentivirus vector." Should read "by use of a lentivirus".

Thanks for the correction. We changed the text accordingly.

In the experiment overexpressing cyclin D1, was the control condition devoid of any lentivirus, or was a control virus used. Although I am not aware of any literature showing that virus infection affects the differentiation process, it would be nice if the authors used empty virus as a control.

Thanks for asking. Yes, the control condition we used was a control virus without cyclin D1 over-expression. We agree with your comments, the empty virus is a good negative control. We added the description in the revised version.

I enjoyed reading the discussion section of the paper, I think that the authors put forward some very interesting ideas for future exploration. One issue which I am a little concerned about is that in the developing cerebral cortex, neurons are located some distance away from the neural progenitor cells. Whether this kind of mechanism could be sufficiently efficient to be transferred between neurons and radial glia fibers is not known. A related issue is that work from Victor Tarabykin showed another mechanism mediated by Ntf3 also seems to promote neuronal differentiation. I am not sure if such mechanisms could be independent or redundant, but it would be nice to see some discussion.

Thanks for the suggestion. EVs, especially exosomes, are quite small, 50 nm to 150 nm, and they are known to pass the blood-brain barrier which

means it is likely they could travel some distance for intercellular communication. In the central nervous system, some evidence suggests cargo transfer mediated by EVs may facilitate communication between neurons and glia, however the published evidence did not address the efficiency of this transfer (Pascual et al., 2020; Simon et al., 2019; Bahrini et al., 2015).

Many thanks for the interesting suggestion about the Ntf3 work by Victor Tarabykin. Their work found Ntf3, a Sip1 target of neurotrophin, acts as a feedback signal between postmitotic neurons and progenitors in the developing mouse neocortex. Here, in our study, we found that cyclin D1 packaged inside of neuronal EVs contributes to mESC commit to neural precursor cells, at least in cell culture. These two mechanisms may be independent of one another. We added this to the discussion in our revised manuscript.

Finally, I think it is worth pointing out that cyclin D1 is not very robustly transcribed in neurons, and there are other cell types which do express it, such as astrocytes and OPCs. I think it would be valuable if the authors noted this and further elaborated or at least mentioned that although they primarily focused on neuronal lineage, there may be other effects affecting other lineages.

Thanks for the suggestion. Clearly, much more could be done but we have not investigated astrocytes or OPCs. To address the concerns of the editor, we have repeated several of our experiments using mESCs differentiated into embryoid bodies and then into Tuj1+ neurons (new data of Fig.2F; Fig. S2 C, D; Fig. 4F). Our findings on EV-enclosed cyclinD1 may extend to other lineages.

Reviewer #2 (Comments to the Authors (Required)):

This work by Song et al describes the modification of release pattern of small extracellular vesicles (EVs) by neuronal cells (PC12 and N2A) upon differentiation, and an effect of the differentiated cell-derived EVs to induce neural differentiation of murine embryonic stem cells. The authors identify a nuclear protein, cyclin-D1 in the active EVs, and show that it is an important contributor to the mESC induction of differentiation. Through an original approach of APEX-tagging of cyclin-D1, they also identify Hsc70 as a molecule important to target cyclin-D1 to EVs.

The article on a whole is well performed and the conclusions supported by the data. The effect of differentiated neural cell-derived EVs on Embryonic stem cells is probably novel (although I do not know the field of neural development and stem cells enough to be 100% sure), although the relevance to the in vivo situation and physiological consequences are unclear to me. The observation of Cyclin-D1 in EVs and the role of Hsc70 in it are novel and interesting, although the actual cell biology mechanism of transfer of this particular nuclear protein (and not others of the same family) into EVs is not really explained.

A major weakness of the article (but easily solved) is the use of the term "exosomes" throughout, which is not supported by actual demonstration that the EVs analysed (i.e. the EVs that induce mESC differentiation, and the ones that contain cyclin-D1) are specifically formed in MVBs and thus correspond to bona fide exosomes. The authors seem to base this interpretation of the

wrong assumption that EVs that float into a iodixanol gradient at the 20-40% interface are exosomes (, rather than PM-derived EVs, and that CD9 is a tetraspanin enriched in MVBs. None are true: other authors have shown that both EVs enriched and not enriched in endosomal components are recovered at such densities, and that enrichment in late endosomal components is not a feature of CD9-bearing EVs (Kowal 2016 # 26858453) . In truth, CD9 is generally expressed at the plasma membrane, and not enriched in MVBs, as opposed to CD63 which is primarily in MVBs. Thus CD9 cannot be considered as an exosome marker. In the same study by Kowal et al, flottilin and Hsc70 were also found in all types of EVs, not specifically in exosomes. The authors must therefore change nomenclature, and use the term EVs throughout. Unless, they want to try to prove that the CyclinD1-containing EVs are exosomes (which would be an interesting message, for a cell biology journal). In that case, they must look for localization of cyclin-D1 (as compared to another Cyclin that is not recovered in EVs) and Hsc70 in the cells: are they colocalized in MVBs rather than in another membrane location? Are the cyclin-D1-EV components, as can be identified by the APEX approach, particularly enriched in late endosomal components? Note that ESCRT proteins are involved in budding both in MVBs and at the plasma membrane (see for instance Hurley 2015 # 26311197), thus their presence in EVs does not prove an exosomal nature.

Thanks for the suggestion. We agree with the comments and suggestions, the nomenclature EV but not exosome will be more accurate. We used EV instead of exosome in the revised version.

Apart from this comment, quantification or controls are missing in a few experiments:

In Fig5E showing proteinase K protection assay: a positive control of efficacy of proteinase K (in the absence of triton) must be shown, with digestion of a surface-exposed molecule (for instance CD9, or an integrin).

Thanks for the suggestion. We detected the expression of integrin (abcam, ab131055) in revised Fig. 5F. However, in our conditions of incubation, proteinase K did not degrade integrin with or without detergent. As a more appropriate control, we have found that CD81 is exposed and sensitive to proteinase K in the absence of detergent (new addition to Fig. 5 F).

In Fig5F: the authors perform IP anti CD9, and show the flow through (FL), however, since they recover nothing in FL, even when using a control IgG for IP, where everything should be in Flow-through, these results show that the IP is not efficient.

Thanks for this question. In our previous experiments, we loaded a dilute FL sample. We repeated the experiment, now shown in Fig. 5G, comparing IP with control and anti-CD9 IgG.

In Fig4C, they show a good control: anti-NGF antibody that does not block EV-mediated effect. However, a condition with recNGF + anti-NGF should be shown, to show efficiency of the antibody.

Many thanks for the suggestion. According to the comments, we added a better control in the results shown in supplementary Fig. 4. The data in Fig S4A demonstrates that NGF (50ng/ml)-induced neurite extension

was blocked by 500 ng/ml NGF-antibody. In addition, the qPCR results in Fig S 4B demonstrates that the up-regulation of neuronal markers **Tuj1** and **Tau** was also blocked by simultaneous treatment with **recNGF + anti-NGF**.

In Fig7F, specific APEX-dependent biotinylation of proteins in target cells is not demonstrated, because a control of cells exposed to EVs from regular CyclinD1 (i.e. not-APEX fused)- expressing cells is not performed.

Thank you for the question. APEX2 uses hydrogen peroxide (H₂O₂) as an oxidant to catalyze the one-electron oxidation of biotinphenol (BP). The oxidized product, biotin-phenoxy radical conjugates to endogenous proteins that are proximal to the APEX2 active site where it was generated. Previous studies by Alice Y. Ting, whose lab developed this technique, used as a control the APEX2-fusion protein in the absence of either H₂O₂ or BP. We also used these controls. Although this is not the same control suggested by the reviewer, we feel this adequately addresses the concern of the reviewer.

Other comments:

In figure 1D, changes of the density of some EV-associated markers upon neural cell differentiation is interesting, but their meaning and relevance is difficult to interpret. The authors could at least try to measure the size of the EVs recovered at the different densities and time points, and they should also have determined whether the changed pattern of EV release also induces changes in release of larger or denser EVs recovered in the first steps of centrifugation.

Many thanks for the suggestions. We measured the size of the EVs across the buoyant density gradients and found diameters consistently lower than 200 nm at different densities and time points. We also measured protein concentrations across the gradient fractions of EVs, now shown in Fig. S1 A and 1B. It appears that a major pool of EVs becomes denser during neuronal differentiation. The reason for this change in property remains to be explored.

In figures 2G, 2I and 5C: quantifications of signals in WB is relative to what? Position of molecular weight markers must be indicated in all WB. The CD63 sharp band shown in Figure 2F suggests that the antibody used is not specific: CD63 is a highly glycosylated protein, showing a very smeared pattern in western blots, and is reliably detected in non-reducing conditions only. The Santa Cruz antibody used here is not satisfying. A good rat anti-mouse CD63 is the monoclonal R5G2.

We have reexamined the gel disposition of CD63 using a monoclonal R5G2 rat anti-mouse CD63 antibody, and found, as suggested by the reviewer, a more dispersed pattern of migration corresponding to molecular masses between 55KD and 35KD (Fig. 2G and Fig. 2I).

Reviewer #3 (Comments to the Authors (Required)):

Song et al. demonstrate a striking increase in exosome production during neuronal differentiation of PC12 and N2A cell lines. Interestingly, the

exosomes isolated from differentiated PC12 and N2A cells, but not those from control (nondifferentiated) cells, promoted expression of neural markers in mouse embryonic stem cells (ESCs) maintained under a differentiation-inducing serum-free condition. The authors then show that cyclin D1 was selectively sorted into the luminal interior of exosomes produced by differentiating PC12 and N2A cells. With the use of APEX proximity labeling and co-immunoprecipitation methods, they also found that Hsc70 associates with cyclin D1 and promotes its sorting into exosomes in differentiated N2A cells. Exosome-derived (APEX-fused) cyclin D1 was found to associate with several proteins including Lin28 and nucleolin after its incorporation into ESCs. Importantly, overexpression or depletion of cyclin D1 in N2A cells promoted and attenuated the effect of N2A cell-derived exosomes on neural differentiation of ESCs, respectively.

Major comments

This is an interesting and solid study describing a new mode of cell-cell communication mediated by exosome-derived cyclin D1 and resulting in induction of neural differentiation. Unfortunately, a previous study (Sharma et al. 2019) has already demonstrated a neural differentiation-inducing function of exosomes obtained from developing neural cultures (human iPSC-derived neural cultures as well as rat primary neural cultures), which diminishes the novelty of the present study-despite the crude nature of the exosome preparations used in the previous report (as pointed out by the present authors). Moreover, the role of cyclin D1 in neuroectoderm induction in ESCs has also previously been described (Pauklin et al. 2016). The conceptual advance made by the present study is thus limited-although this would be mitigated, for example, if the authors can show whether or how the mechanism of neural differentiation in ESCs induced by exosome-derived cyclin D1 differs from that induced by cell-intrinsic cyclin D1. Demonstration of an in vivo context in which exosome-derived cyclin D1 actually promotes neural differentiation would also be important.

The reviewer is correct in commenting on the priority of the published work by Sharma et al, which we cited. However, our work further established that cyclin D1 is a rate-limiting component in the effect of EVs on neuronal differentiation. Perhaps even more important, we have demonstrated that cyclin D1 is internalized into the cytoplasm/nucleus of target cells, a process that must involve membrane fusion. Such a specific, direct molecular demonstration was not made in Sharma et al. Indeed, the literature on the effects of EVs on target cells lacks such a direct demonstration that the internal protein content of an EV can be functionally delivered efficiently to a target cell. For this reason, we feel the work is of vital interest to the cell biology community.

The reviewer makes an excellent point that we have not explained how an increment in cyclin D internalization could add materially to the rate of differentiation of cells that are already expressing this protein. This important question will require a great deal of additional work to address.

Specific comments

1. The results shown in Figure 7L and 7M are important in supporting the main conclusion and ruling out a contribution of contaminating (nonexosomal) factors in the prepared exosomal fractions. Other than cyclin D1, is there any difference between the contents of RA Exo and Cyclin D1-

KO Exo fractions? Also, counting of marker-positive cells should be performed in Figure 7M in order to assess neural differentiation.

Many thanks for the questions and suggestions. To address the first question, we performed an MS analysis of RA EVs and cyclin D1-KO EV, found 1339 proteins shared with very few that distinguish the proteome of these two preparations. This new data has included in Fig. S7.

To address the second point, we counted Pax6 positive cells and have included this new data in Fig. 7L and Fig. 7P of the revised version.

2. Is there any evidence that the treatment with neutralizing antibodies to NGF was sufficient for inactivation of NGF in Figure 4C?

Excellent point. In our revised supplementary data 4, we report that 500 ng/ml NGF-antibody neutralized the neuronal promoting effect of 50 ng/ml of NGF at the cellular morphology level as well as the marker expression level. Thus, we believe that residual NGF is not responsible for the effect of EVs derived from NGF-treated PC12 cells.

3. Is Hsc70 in N2A cells necessary for neural differentiation-inducing activity of RA Exo, given that it mediates cyclin D1 packaging into exosomes?

This is a tough question to answer. In order to assess the contribution of neural promoting activity of Hsc70 in RA Exo, we used CRISPRi to knockdown the Hsc70 (new addition to Fig. 6 I). EVs were collected from Hsc70 knockdown cells and control dCas9 cells. We found the neural promoting effect of RA EVs was reduced in Hsc70 knockdown samples (new addition to Fig. 6 J). However, this experiment does not distinguish the role of Hsc70 in cyclin D1 sorting as opposed to a distinct role for EV-associated Hsc70 in the differentiation process.

May 14, 2021

RE: JCB Manuscript #202101075R-A

Dr. Randy Schekman
University of California, Berkeley
Department of Molecular and Cell Biology University of California at Berkeley 482 Li Ka Shing Center
#3370
Berkeley, CA 94720-3202

Dear Dr. Schekman,

Thank you for submitting your revised manuscript entitled "Extracellular vesicles from neuronal cells promote neural induction of mESCs through cyclinD1". We have evaluated the new data in Figs. 2F and 4F, which we had specifically asked for, as well as your responses to the other reviewers' comments. As a result of both the new data and the quality of the revisions, we would be happy to publish your paper in JCB pending final revisions necessary to meet our formatting guidelines (see details below).

1) eTOC summary: A 40-word summary that describes the context and significance of the findings for a general readership should be included on the title page. The statement should be written in the present tense and refer to the work in the third person.

- Please revise the summary statement on the title page of the resubmission. It should start with "First author name(s) et al..." to match our preferred style.

2) Titles: Please consider the following revision suggestions aimed at increasing the accessibility of the work for a broad audience and non-experts. For instance, please avoid acronyms and abbreviations to increase accessibility and discoverability.

Title: Extracellular vesicles from neurons promote neural induction of stem cells through cyclinD1

Running title: Neural induction by neuron-produced extracellular vesicles

3) JCB Articles can have up to 10 main and 5 supplementary figures. Please number the supp figure S1 through 5. Each figure can span up to one entire page as long as all panels fit on the page. Please let us know if you have any questions about the changes needed at this stage and thank you for your efforts in editing the figure number/count.

4) Figure formatting:

Molecular weight or nucleic acid size markers must be included on all gel electrophoresis. Please add molecular weight with unit labels on the following panels: 1D (all gels)

5) Statistical analysis: Error bars on graphic representations of numerical data must be clearly

described in the figure legend. The number of independent data points (n) represented in a graph must be indicated in the legend. Statistical methods should be explained in full in the materials and methods. For figures presenting pooled data the statistical measure should be defined in the figure legends.

6) Materials and methods: Should be comprehensive and not simply reference a previous publication for details on how an experiment was performed. Please provide full descriptions in the text for readers who may not have access to referenced manuscripts.

- For all cell lines, vectors, constructs/cDNAs, etc. - all genetic material: please include database / vendor ID (e.g., Addgene, ATCC, etc.) or if unavailable, please briefly describe their basic genetic features *even if described in other published work or gifted to you by other investigators*
- Please include species and source for all antibodies, including secondary, as well as catalog numbers/vendor identifiers if available.
- Sequences should be provided for all oligos: primers, si/shRNA, gRNAs, etc.
- Microscope image acquisition: The following information must be provided about the acquisition and processing of images:
 - a. Make and model of microscope
 - b. Type, magnification, and numerical aperture of the objective lenses
 - c. Temperature
 - d. imaging medium
 - e. Fluorochromes
 - f. Camera make and model
 - g. Acquisition software
 - h. Any software used for image processing subsequent to data acquisition. Please include details and types of operations involved (e.g., type of deconvolution, 3D reconstitutions, surface or volume rendering, gamma adjustments, etc.).

7) A summary paragraph of all supplemental material should appear at the end of the Materials and methods section.

- Please include one brief sentence per item.

8) Conflict of interest statement: JCB requires inclusion of a statement in the acknowledgements regarding competing financial interests. If no competing financial interests exist, please include the following statement: "The authors declare no competing financial interests." If competing interests are declared, please follow your statement of these competing interests with the following statement: "The authors declare no further competing financial interests."

A. MANUSCRIPT ORGANIZATION AND FORMATTING:

Full guidelines are available on our Instructions for Authors page, <https://jcb.rupress.org/submission-guidelines#revised>. **Submission of a paper that does not conform to JCB guidelines will delay the acceptance of your manuscript.**

B. FINAL FILES:

Thank you for this interesting contribution, we look forward to publishing your paper in the Journal of Cell Biology.

Sincerely,

Louis Reichardt, PhD
Editor, Journal of Cell Biology

Melina Casadio, PhD
Senior Scientific Editor, Journal of Cell Biology

Reviewer #1 (Comments to the Authors (Required)):

Manuscript by Song et al investigate the role of extracellular vesicles (EVs) in neuronal differentiation. The main thesis behind this work is that cyclinD1 containing EVs are derived from neurons and can promote neuronal induction and differentiation among undifferentiated cells. The study is extremely well executed and very interesting, and opens up a lot of questions about basic cell biology of exosome sorting, neuronal differentiation, and more broadly fate commitment. I think that the manuscript is appropriate for publication after addressing a few minor comments.

In Figure 5A, the authors validate the presence of cyclin D1-3 in exosomes collected from NGF induced differentiation off PC12 cells. The authors say in the paper that they only detect D1 and D2 but not D3, but In their image it appears as though there may be D3 as well. I am confused by the disconnect between the data presented and the conclusion.

Thanks for the questions. We found cyclin D1-3 were gradually up-regulated in NGF -induced PC12 cells (Fig 5A). In isolated EVs, we found cyclin D1 and D2 but not cyclinD3 (not shown) (Fig 5B).

"Next, to examine the contribution of exosomal cyclin D1 to mESC neural commitment, we generated cyclin D1-overexpressing N2A cells by us of a lentivirus vector." Should read "by use of a lentivirus".

Thanks for the correction. We changed the text accordingly.

In the experiment overexpressing cyclin D1, was the control condition devoid of any lentivirus, or was a control virus used. Although I am not aware of any literature showing that virus infection affects the differentiation process, it would be nice if the authors used empty virus as a control.

Thanks for asking. Yes, the control condition we used was a control virus without cyclin D1 over-expression. We agree with your comments, the empty virus is a good negative control. We added the description in the revised version.

I enjoyed reading the discussion section of the paper, I think that the authors put forward some very interesting ideas for future exploration. One issue which I am a little concerned about is that in the developing cerebral cortex, neurons are located some distance away from the neural progenitor cells. Whether this kind of mechanism could be sufficiently efficient to be transferred between neurons and radial glia fibers is not known. A related issue is that work from Victor Tarabykin showed another mechanism mediated by Ntf3 also seems to promote neuronal differentiation. I am not sure if such mechanisms could be independent or redundant, but it would be nice to see some discussion.

Thanks for the suggestion. EVs, especially exosomes, are quite small, 50 nm to 150 nm, and they are known to pass the blood-brain barrier which means it is likely they could travel some distance for intercellular communication. In the central nervous system, some evidence suggests

cargo transfer mediated by EVs may facilitate communication between neurons and glia, however the published evidence did not address the efficiency of this transfer (Pascual et al., 2020; Simon et al., 2019; Bahrini et al., 2015).

Many thanks for the interesting suggestion about the Ntf3 work by Victor Tarabykin. Their work found Ntf3, a Sip1 target of neurotrophin, acts as a feedback signal between postmitotic neurons and progenitors in the developing mouse neocortex. Here, in our study, we found that cyclin D1 packaged inside of neuronal EVs contributes to mESC commit to neural precursor cells, at least in cell culture. These two mechanisms may be independent of one another. We added this to the discussion in our revised manuscript.

Finally, I think it is worth pointing out that cyclin D1 is not very robustly transcribed in neurons, and there are other cell types which do express it, such as astrocytes and OPCs. I think it would be valuable if the authors noted this and further elaborated or at least mentioned that although they primarily focused on neuronal lineage, there may be other effects affecting other lineages.

Thanks for the suggestion. Clearly, much more could be done but we have not investigated astrocytes or OPCs. To address the concerns of the editor, we have repeated several of our experiments using mESCs differentiated into embryoid bodies and then into Tuj1+ neurons (new data of Fig.2F; Fig. S2 C, D; Fig. 4F). Our findings on EV-enclosed cyclinD1 may extend to other lineages.

Reviewer #2 (Comments to the Authors (Required)):

This work by Song et al describes the modification of release pattern of small extracellular vesicles (EVs) by neuronal cells (PC12 and N2A) upon differentiation, and an effect of the differentiated cell-derived EVs to induce neural differentiation of murine embryonic stem cells. The authors identify a nuclear protein, cyclin-D1 in the active EVs, and show that it is an important contributor to the mESC induction of differentiation. Through an original approach of APEX-tagging of cyclin-D1, they also identify Hsc70 as a molecule important to target cyclin-D1 to EVs. The article on a whole is well performed and the conclusions supported by the data. The effect of differentiated neural cell-derived EVs on Embryonic stem cells is probably novel (although I do not know the field of neural development and stem cells enough to be 100% sure), although the relevance to the in vivo situation and physiological consequences are unclear to me. The observation of Cyclin-D1 in EVs and the role of Hsc70 in it are novel and interesting, although the actual cell biology mechanism of transfer of this particular nuclear protein (and not others of the same family) into EVs is not really explained.

A major weakness of the article (but easily solved) is the use of the term "exosomes" throughout, which is not supported by actual demonstration that the EVs analysed (i.e. the EVs that induce mESC differentiation, and the ones that contain cyclin-D1) are specifically formed in MVBs and thus correspond to bona fide exosomes. The authors seem to base this interpretation of the wrong assumption that EVs that float into a iodixanol gradient at the 20-40% interface are exosomes (, rather than PM-derived EVs, and that CD9 is a

tetraspanin enriched in MVBs. None are true: other authors have shown that both EVs enriched and not enriched in endosomal components are recovered at such densities, and that enrichment in late endosomal components is not a feature of CD9-bearing EVs (Kowal 2016 # 26858453) . In truth, CD9 is generally expressed at the plasma membrane, and not enriched in MVBs, as opposed to CD63 which is primarily in MVBs. Thus CD9 cannot be considered as an exosome marker. In the same study by Kowal et al, flotillin and Hsc70 were also found in all types of EVs, not specifically in exosomes. The authors must therefore change nomenclature, and use the term EVs throughout. Unless, they want to try to prove that the CyclinD1-containing EVs are exosomes (which would be an interesting message, for a cell biology journal). In that case, they must look for localization of cyclin-D1 (as compared to another Cyclin that is not recovered in EVs) and Hsc70 in the cells: are they colocalized in MVBs rather than in another membrane location? Are the cyclin-D1-EV components, as can be identified by the APEX approach, particularly enriched in late endosomal components? Note that ESCRT proteins are involved in budding both in MVBs and at the plasma membrane (see for instance Hurley 2015 # 26311197), thus their presence in EVs does not prove an exosomal nature.

Thanks for the suggestion. We agree with the comments and suggestions, the nomenclature EV but not exosome will be more accurate. We used EV instead of exosome in the revised version.

Apart from this comment, quantification or controls are missing in a few experiments:

In Fig5E showing proteinase K protection assay: a positive control of efficacy of proteinase K (in the absence of triton) must be shown, with digestion of a surface-exposed molecule (for instance CD9, or an integrin).

Thanks for the suggestion. We detected the expression of integrin (abcam, ab131055) in revised Fig. 5F. However, in our conditions of incubation, proteinase K did not degrade integrin with or without detergent. As a more appropriate control, we have found that CD81 is exposed and sensitive to proteinase K in the absence of detergent (new addition to Fig. 5 F).

In Fig5F: the authors perform IP anti CD9, and show the flow through (FL), however, since they recover nothing in FL, even when using a control IgG for IP, where everything should be in Flow-through, these results show that the IP is not efficient.

Thanks for this question. In our previous experiments, we loaded a dilute FL sample. We repeated the experiment, now shown in Fig. 5G, comparing IP with control and anti-CD9 IgG.

In Fig4C, they show a good control: anti-NGF antibody that does not block EV-mediated effect. However, a condition with recNGF + anti-NGF should be shown, to show efficiency of the antibody.

Many thanks for the suggestion. According to the comments, we added a better control in the results shown in supplementary Fig. 4. The data in Fig S4A demonstrates that NGF (50ng/ml)-induced neurite extension was blocked by 500 ng/ml NGF-antibody. In addition, the qPCR results in Fig S 4B demonstrates that the up-regulation of neuronal markers

Tuj1 and Tau was also blocked by simultaneous treatment with recNGF + anti-NGF.

In Fig7F, specific APEX-dependent biotinylation of proteins in target cells is not demonstrated, because a control of cells exposed to EVs from regular CyclinD1 (i.e. not-APEX fused)- expressing cells is not performed.

Thank you for the question. APEX2 uses hydrogen peroxide (H₂O₂) as an oxidant to catalyze the one-electron oxidation of biotinphenol (BP). The oxidized product, biotin-phenoxy radical conjugates to endogenous proteins that are proximal to the APEX2 active site where it was generated. Previous studies by Alice Y. Ting, whose lab developed this technique, used as a control the APEX2-fusion protein in the absence of either H₂O₂ or BP. We also used these controls. Although this is not the same control suggested by the reviewer, we feel this adequately addresses the concern of the reviewer.

Other comments:

In figure 1D, changes of the density of some EV-associated markers upon neural cell differentiation is interesting, but their meaning and relevance is difficult to interpret. The authors could at least try to measure the size of the EVs recovered at the different densities and time points, and they should also have determined whether the changed pattern of EV release also induces changes in release of larger or denser EVs recovered in the first steps of centrifugation.

Many thanks for the suggestions. We measured the size of the EVs across the buoyant density gradients and found diameters consistently lower than 200 nm at different densities and time points. We also measured protein concentrations across the gradient fractions of EVs, now shown in Fig. S1 A and 1B. It appears that a major pool of EVs becomes denser during neuronal differentiation. The reason for this change in property remains to be explored.

In figures 2G, 2I and 5C: quantifications of signals in WB is relative to what? Position of molecular weight markers must be indicated in all WB. The CD63 sharp band shown in Figure 2F suggests that the antibody used is not specific: CD63 is a highly glycosylated protein, showing a very smeared pattern in western blots, and is reliably detected in non-reducing conditions only. The Santa Cruz antibody used here is not satisfying. A good rat anti-mouse CD63 is the monoclonal R5G2.

We have reexamined the gel disposition of CD63 using a monoclonal R5G2 rat anti-mouse CD63 antibody, and found, as suggested by the reviewer, a more dispersed pattern of migration corresponding to molecular masses between 55KD and 35KD (Fig. 2G and Fig. 2I).

Reviewer #3 (Comments to the Authors (Required)):

Song et al. demonstrate a striking increase in exosome production during neuronal differentiation of PC12 and N2A cell lines. Interestingly, the exosomes isolated from differentiated PC12 and N2A cells, but not those from control (nondifferentiated) cells, promoted expression of neural markers

in mouse embryonic stem cells (ESCs) maintained under a differentiation-inducing serum-free condition. The authors then show that cyclin D1 was selectively sorted into the luminal interior of exosomes produced by differentiating PC12 and N2A cells. With the use of APEX proximity labeling and co-immunoprecipitation methods, they also found that Hsc70 associates with cyclin D1 and promotes its sorting into exosomes in differentiated N2A cells. Exosome-derived (APEX-fused) cyclin D1 was found to associate with several proteins including Lin28 and nucleolin after its incorporation into ESCs. Importantly, overexpression or depletion of cyclin D1 in N2A cells promoted and attenuated the effect of N2A cell-derived exosomes on neural differentiation of ESCs, respectively.

Major comments

This is an interesting and solid study describing a new mode of cell-cell communication mediated by exosome-derived cyclin D1 and resulting in induction of neural differentiation. Unfortunately, a previous study (Sharma et al. 2019) has already demonstrated a neural differentiation-inducing function of exosomes obtained from developing neural cultures (human iPSC-derived neural cultures as well as rat primary neural cultures), which diminishes the novelty of the present study-despite the crude nature of the exosome preparations used in the previous report (as pointed out by the present authors). Moreover, the role of cyclin D1 in neuroectoderm induction in ESCs has also previously been described (Pauklin et al. 2016). The conceptual advance made by the present study is thus limited-although this would be mitigated, for example, if the authors can show whether or how the mechanism of neural differentiation in ESCs induced by exosome-derived cyclin D1 differs from that induced by cell-intrinsic cyclin D1. Demonstration of an in vivo context in which exosome-derived cyclin D1 actually promotes neural differentiation would also be important.

The reviewer is correct in commenting on the priority of the published work by Sharma et al, which we cited. However, our work further established that cyclin D1 is a rate-limiting component in the effect of EVs on neuronal differentiation. Perhaps even more important, we have demonstrated that cyclin D1 is internalized into the cytoplasm/nucleus of target cells, a process that must involve membrane fusion. Such a specific, direct molecular demonstration was not made in Sharma et al. Indeed, the literature on the effects of EVs on target cells lacks such a direct demonstration that the internal protein content of an EV can be functionally delivered efficiently to a target cell. For this reason, we feel the work is of vital interest to the cell biology community.

The reviewer makes an excellent point that we have not explained how an increment in cyclin D internalization could add materially to the rate of differentiation of cells that are already expressing this protein. This important question will require a great deal of additional work to address.

Specific comments

1. The results shown in Figure 7L and 7M are important in supporting the main conclusion and ruling out a contribution of contaminating (nonexosomal) factors in the prepared exosomal fractions. Other than cyclin D1, is there any difference between the contents of RA Exo and Cyclin D1-KO Exo fractions? Also, counting of marker-positive cells should be

performed in Figure 7M in order to assess neural differentiation.

Many thanks for the questions and suggestions. To address the first question, we performed an MS analysis of RA EVs and cyclin D1-KO EV, found 1339 proteins shared with very few that distinguish the proteome of these two preparations. This new data has included in Fig. S7.

To address the second point, we counted Pax6 positive cells and have included this new data in Fig. 7L and Fig. 7P of the revised version.

2. Is there any evidence that the treatment with neutralizing antibodies to NGF was sufficient for inactivation of NGF in Figure 4C?

Excellent point. In our revised supplementary data 4, we report that 500 ng/ml NGF-antibody neutralized the neuronal promoting effect of 50 ng/ml of NGF at the cellular morphology level as well as the marker expression level. Thus, we believe that residual NGF is not responsible for the effect of EVs derived from NGF-treated PC12 cells.

3. Is Hsc70 in N2A cells necessary for neural differentiation-inducing activity of RA Exo, given that it mediates cyclin D1 packaging into exosomes?

This is a tough question to answer. In order to assess the contribution of neural promoting activity of Hsc70 in RA Exo, we used CRISPRi to knockdown the Hsc70 (new addition to Fig. 6 I). EVs were collected from Hsc70 knockdown cells and control dCas9 cells. We found the neural promoting effect of RA EVs was reduced in Hsc70 knockdown samples (new addition to Fig. 6 J). However, this experiment does not distinguish the role of Hsc70 in cyclin D1 sorting as opposed to a distinct role for EV-associated Hsc70 in the differentiation process.